# Integrating apaQTL and eQTL analysis identifies a potential causal variant associated with lung adenocarcinoma risk in the Chinese population
Huiwen Xu[1,5], Yutong Wu[1,5], Qiong Chen[1,5], Yuhui Yu[2,3,5], Qianyao Meng[4], Na Qin[2,3], Wendi Zhang[1], Xiaobo Tao[1], Siqi Li[1], Tian Tian[1], Lei Zhang[1], Hongxia Ma[2,3], Jiahua Cui ☉ [1] ✉ & Minjie Chu ☉ [1] ✉

Alternative polyadenylation (APA) plays a crucial role in cancer biology. Here, we used data from the 3′ aQTL-atlas, GTEx, and the China Nanjing Lung Cancer GWAS database to explore the association between apaQTL/eQTL-SNPs and the risk of lung adenocarcinoma (LUAD). The variant T allele of rs277646 in *NIT2* is associated with an increased risk of LUAD (OR = 1.12, *P* = 0.015), lower PDUI values, and higher *NIT2* expression. The 3′RACE experiment showed multiple poly (A) sites in *NIT2*, with the rs277646-T allele causing preferential use of the proximal poly (A) site, resulting in a shorter 3′ UTR transcript. This leads to the loss of the hsa-miR-650 binding site, thereby affecting LUAD malignant phenotypes by regulating the expression level of *NIT2*. Our findings may provide new insights into understanding and exploring APA events in LUAD carcinogenesis.

Lung adenocarcinoma (LUAD) and lung squamous cell carcinoma (LUSC) have shown distinct incidence trends in recent decades[1,2]. In China, the incidence of LUSC showed a decreasing trend over time, while the incidence of LUAD showed a significantly increasing trend[3]. The correlation between LUAD and smoking is weaker than that of LUSC[4], and it is more closely related to other non-smoking (cooking fumes, PM$_{2.5}$, et al.) and genetic factors[5,6]. The evidence from the GWAS Catalog (https://www.ebi.ac.uk/gwas/) showed that the currently reported LUAD susceptibility regions are far more than that of LUSC, indicating that genetic factors may affect the susceptibility of LUAD in a large extent. Therefore, screening of genetically susceptible high-risk individuals and taking targeted preventive measures may be one of the strategies to reduce the morbidity and mortality of LUAD.

Genome-wide association studies (GWAS) have identified plenty of single nucleotide polymorphisms (SNPs) associated with LUAD[7-9]. However, most of the loci discovered by GWAS are located in noncoding regions, and elucidating the molecular mechanisms of which are difficult. Quantitative trait locus (QTL) mapping, such as eQTL analysis, is one of the most common research strategies to assess the biological function of SNPs, and has been demonstrated as a powerful tool to clarify the relationship between SNPs and gene expression[10]. Although a few noncoding SNPs function as eQTLs, the function of most disease-related non-coding SNPs are not yet clear. Altogether, eQTL analysis can only provide clues to the correlation between SNPs and gene expression, and is not yet effective in revealing the mediating mechanisms on how different genotypes of SNPs regulate abnormal gene expression.

Polyadenylation is an important post-transcriptional regulatory mechanism, and with the rapid development of transcriptomic, it has been found that alternative polyadenylation (APA) is widespread in more than 70% of human genes[11], which may produce mRNA isoforms with different lengths of 3′untranslated regions (3′UTR)[12], hence enable mRNA isoforms to have different stability, cellular localization, and translational efficiency[13]. Recently, Xiang et al. analyzed APA events in 17 different tumor tissues and found that APA events were detected at a higher frequency in LUAD tissues, ranking 4th[14]. Therefore, exploring APA-related genes in LUAD and searching regulators of which may provide novel clues to investigate the pathogenesis of LUAD.

SNP as a regulator of APA events may mediate the differential selection of poly(A) sites of related genes, leading to differences in cleavage sites and then produce transcripts with different 3′UTR lengths[15]. The functional

[1]Department of Epidemiology, School of Public Health, Nantong University, Nantong, Jiangsu, China. [2]Department of Epidemiology, Center for Global Health, School of Public Health, Nanjing Medical University, Nanjing, Jiangsu, China. [3]Jiangsu Key Lab of Cancer Biomarkers, Prevention and Treatment, Collaborative Innovation Center for Cancer Personalized Medicine, Nanjing Medical University, Nanjing, Jiangsu, China. [4]Department of Global Health and Population, School of Public Health, Harvard University, Boston, MA, USA. [5]These authors contributed equally: Huiwen Xu, Yutong Wu, Qiong Chen, Yuhui Yu. ✉e-mail: cuijiahua@ntu.edu.cn; chuminjie@ntu.edu.cn

SNPs associated with APA events can be used as a new tool to discover the regulatory factors of APA events, and will provide an important theoretical basis for interpreting the functional SNPs in the non-coding region. Recently, Cui et al. have obtained 473,008 functional quantitative trait loci associated with APA events (apaQTL-SNPs) in lung tissue by DaPars2 bioinformatics algorithm[16] using RNA-seq data from the GTEx database[17]. However, the above lung tissue-related apaQTL-SNPs data were obtained by bioinformatics analysis, not linking the discovered apaQTL-SNPs to APA-related LUAD genes, and did not investigate whether those apaQTL-SNPs may affect the risk of developing LUAD and the extent of their association through large-sample case-control studies.

In this study, to explore the association between apaQTL/eQTL-SNPs and the risk of LUAD, firstly, we obtained APA-related genes in LUAD that with significant correlation between APA events and corresponding gene expression. On this basis, we further screened APA-related LUAD genes that with consistently differential expression at both the mRNA level and its' coding protein level by integrating transcriptomics and proteomics data. Then, the 3′aQTL-atlas database was used to screen candidate apaQTL-SNPs that were located on the above APA-related LUAD genes. Subsequently, we combined eQTL analysis and obtained SNPs with both eQTL and apaQTL functions. Then, we used genome-wide genetic analysis of large samples (China Nanjing Lung Cancer GWAS database: 8762 LUAD cases and 13,328 healthy controls) to explore the association between candidate apaQTL/eQTL-SNPs and LUAD risk. Finally, functional experiments were performed to illustrate the effects of the identified apaQTL/eQTL-SNP on the malignant phenotype of LUAD in vitro and in vivo, while the occurrence of APA events under different alleles of the identified apaQTL/eQTL-SNP was evaluated through 3′RACE technology.

## Results

### Identification of APA-related LUAD genes
A total of 518 APA-related genes in LUAD ($|Rs| > 0.3$, $P_{FDR} < 0.05$) were obtained. Among these, the PDUI values of 285 genes were positively correlated with the corresponding gene expression ($Rs > 0.3$, $P_{FDR} < 0.05$), and the PDUI values of the other 233 genes were negatively correlated with the corresponding gene expression ($Rs < -0.3$, $P_{FDR} < 0.05$).

We further analyzed the expression differences of the above 518 genes at mRNA level between 57 paired LUAD tumor tissues and adjacent non-tumor tissues using the TCGA database. The result showed that a total of 143 genes (mRNA level) were differentially expressed ($|FC| > 1.5$, $P < 0.05$). Then, we validated those 143 genes between 49 paired LUAD tumor tissues and adjacent non-tumor tissues from the Chinese population, and 65 genes were validated as significantly differential expressed ($|FC| > 1.5$, $P < 0.05$).

Based on these validated 65 genes at the mRNA level, we further evaluated whether they were differentially expressed at the level of their corresponding coding proteins.

The results showed that there were 32 proteins differentially expressed between the tumor tissues and adjacent non-tumor tissues from the same Chinese population and the expression directions of 32 proteins were consistent with the corresponding mRNAs.

Thus, the overlapped 32 APA-related LUAD genes with consistent differential expression both at the mRNA and its' coding protein levels were used for further study.

### Identification of apaQTL/eQTL-SNPs in APA-related LUAD genes
A total of 423 apaQTL-SNPs located in the above 32 gene regions ($P_{FDR} < 0.05$) were obtained through a public 3′aQTL-atlas website, while 338 apaQTL-SNPs also showed eQTL functions and may influence expression levels of 9 APA-related LUAD genes ($P < 0.05$). Subsequently, 256 apaQTL/eQTL-SNPs were survived with MAF > 0.05 in CHB. Finally, 28 apaQTL/eQTL-SNPs were selected after LD filter ($r^2 < 0.8$) (Fig. 1) and their detailed information was shown in Table 1.

### The association between candidate apaQTL/eQTL-SNPs and LUAD risk
We further examined the association between 28 apaQTL/eQTL-SNPs and LUAD risk in the Chinese population using the China Nanjing Lung Cancer GWAS database. As shown in Table 2, the variant T allele of rs10452178 located in *CISD2* was significantly associated with a decreased risk of LUAD (OR = 0.92, 95% CI = 0.87–0.98, P = 0.009). The variant T allele of rs277646 located in *NIT2* was significantly associated with an increased risk of LUAD (OR = 1.12, 95% CI = 1.02–1.22, P = 0.015). Besides, the variant T allele of rs11714045 located in *NIT2* showed a borderline significant association with an increased risk of LUAD (OR = 1.05, 95% CI = 1.00–1.10, P = 0.076).

### The correlation between PDUI value and corresponding gene expression
To analyze the APA usage of *CISD2* and *NIT2* in LUAD, we combined PDUI data and gene expression data from TCGA. As shown in Fig. 2a–c, in adjacent non-tumor tissues ($Rs = -0.540$, $P = 1.23 \times 10^{-5}$), LUAD tumor tissues ($Rs = -0.407$, $P = 9.50 \times 10^{-21}$), and total tissues ($Rs = -0.500$, $P = 1.28 \times 10^{-35}$), we all observed a negative correlation between the PDUI value and gene expression level of *CISD2*. As the PDUI value increased, the expression of *CISD2* decreased. In addition, the PDUI value of *NIT2* was also negatively correlated with *NIT2* gene expression level in both adjacent non-tumor tissues ($Rs = -0.371$, $P = 4.12 \times 10^{-3}$), LUAD tumor tissues ($Rs = -0.320$, $P = 5.60 \times 10^{-13}$), and total tissues ($Rs = -0.361$, $P = 3.82 \times 10^{-18}$) (Fig. 2d–f).

### APA analysis of 3 identified apaQTL/eQTL-SNPs
As shown in the Fig. 2g–i, two apaQTL/eQTL-SNPs (rs10452178 and rs11714045) had significantly higher PDUI values under the variant alleles ($P = 4.50 \times 10^{-5}$, $P = 1.58 \times 10^{-20}$), while the other apaQTL/eQTL-SNP (rs277646) had significantly lower PDUI values under the variant T allele ($P = 1.46 \times 10^{-4}$).

### eQTL analysis of 3 apaQTL/eQTL-SNPs
The eQTL analysis indicated significantly lower expression levels of *CISD2* under the variant T alleles of rs10452178 ($P = 4.4 \times 10^{-6}$), and similarly, significantly lower expression levels of *NIT2* was observed under the variant T alleles of rs11714045 ($P = 1.8 \times 10^{-6}$). Besides, significantly higher expression levels of *NIT2* was observed under the variant T alleles of rs277646 ($P = 5.5 \times 10^{-9}$) (Fig. 2j–l).

### Expression analysis of CISD2 and NIT2
The mRNA expression level of *CISD2* ($P = 2.07 \times 10^{-12}$) (Fig. 3a) and *NIT2* ($P = 1.23 \times 10^{-7}$) (Fig. 3b) were both significantly higher in the LUAD tumor tissues ($n = 57$) from TCGA database. We further validated the mRNA expression level of *CISD2* and *NIT2* in the Chinese population. The result showed that the mRNA expression level of *CISD2* and *NIT2* were also higher in LUAD tumor tissues ($n = 49$) compared with paired adjacent non-tumor tissues ($P = 4.31 \times 10^{-3}$, $P = 7.38 \times 10^{-6}$) (Fig. 3c, d). In addition, their corresponding coding proteins also showed the same trend ($P = 4.80 \times 10^{-21}$, $P = 3.31 \times 10^{-23}$), which were up-regulated in LUAD tumor tissues (Fig. 3e, f).

### PDUI analysis of CISD2 and NIT2
Using TCGA database, the PDUI value of *CISD2* and *NIT2* were significantly lower in total LUAD tumor tissues compared with adjacent non-tumor tissues ($P = 2.01 \times 10^{-55}$, $P = 2.90 \times 10^{-7}$) (Fig. 3g, h). We further performed PDUI analysis of *CISD2* and *NIT2* in 56 paired samples, and the results also showed that the PDUI value of *CISD2* and *NIT2* were also significantly lower in LUAD tumor tissues ($P = 8.80 \times 10^{-17}$, $P = 2.88 \times 10^{-5}$) (Fig. 3i, j). This implies that the 3′UTR length of *CISD2* and *NIT2* were significantly shorter in LUAD tumor tissues compared with adjacent non-tumor tissues.

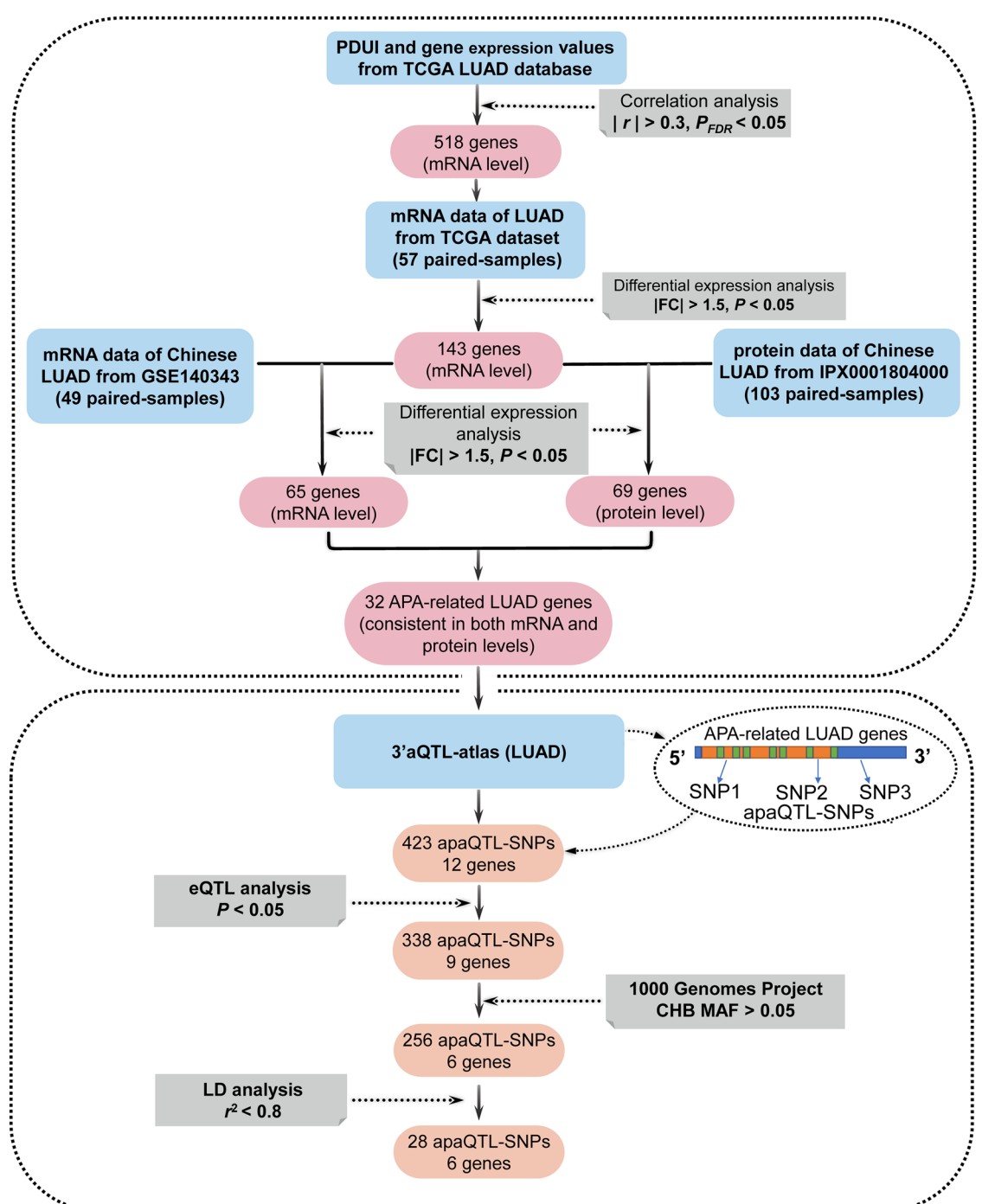

**Fig. 1 | Schematic representation the selection of apaQTL/eQTL-SNPs.** apaQTL alternative polyadenylation quantitative trait loci, SNPs single nucleotide polymorphisms, LUAD lung adenocarcinoma, MAF minor allele frequency, CHB Chinese Han population in Beijing, LD linkage disequilibrium.

## Survival analysis of CISD2 and NIT2

To determine whether the occurrence of APA events in this study was related to the survival of LUAD patients, we performed survival analyses for patients from the TCGA dataset. The results showed that the overall LUAD survival is significantly longer in patients with high PDUI values of *CISD2* and *NIT2* than that with low PDUI values ($P = 7.70 \times 10^{-4}$, $P = 0.033$) (Fig. 3k, l).

## The expression of CISD2 and NIT2 in cell lines

To determine the expression of CISD2 and NIT2 in LUAD cell lines, we performed protein blotting on LUAD cell lines and normal HBE cell line. The results indicate that NIT2 expression in LUAD cell lines (PC9 and

SPCA1) is significantly higher than in HBE cells. Additionally, there is a trend of higher expression of NIT2 in the LUAD cell line A549 compared to HBE cells. However, the expression trend of CISD2 in LUAD cell lines (A549, PC9, and SPCA1) is not consistent (Fig. 4a).

## Effects of NIT2-rs277646 on the malignant phenotype of LUAD in vitro and vivo

Through CRISPR/Cas9 mediated genome editing, we successfully obtained *NIT2*-rs277646-G and performed DNA sequencing for validation (Fig. 4b). Meanwhile, Western blot analysis revealed that the expression of NIT2 in the *NIT2*-rs277646-T was higher than that in the *NIT2*-rs277646-G ($P = 0.0180$) (Fig. 4c). Compared with the *NIT2*-rs277646-G group, there

**Table 1 | Detail information of the 28 candidate apaQTL/eQTL-SNPs**

| No. | SNP | Location | Gene | Region | eQTL-*P* value | apaQTL-*P* value |
|-----|-----|----------|------|--------|----------------|-------------------|
| 1 | rs11771259 | chr7:7237584 | *C1GALT1* | intron | $1.20 \times 10^{-29}$ | $2.50 \times 10^{-11}$ |
| 2 | rs61047106 | chr7:7246226 | *C1GALT1* | 3'UTR | $6.70 \times 10^{-11}$ | $8.58 \times 10^{-06}$ |
| 3 | rs13226913 | chr7:7207215 | *C1GALT1* | intron | $4.60 \times 10^{-19}$ | $1.75 \times 10^{-08}$ |
| 4 | rs1008898 | chr7:7233928 | *C1GALT1* | intron | $2.10 \times 10^{-13}$ | $5.07 \times 10^{-08}$ |
| 5 | rs10278474 | chr7:7229792 | *C1GALT1* | intron | $1.40 \times 10^{-03}$ | $7.26 \times 10^{-12}$ |
| 6 | rs4141099 | chr7:7237321 | *C1GALT1* | intron | $6.30 \times 10^{-16}$ | $5.65 \times 10^{-07}$ |
| 7 | rs75439000 | chr7:7201476 | *C1GALT1* | intron | $6.70 \times 10^{-18}$ | $2.93 \times 10^{-09}$ |
| 8 | rs2108788 | chr7:7219270 | *C1GALT1* | intron | $2.50 \times 10^{-21}$ | $4.10 \times 10^{-11}$ |
| 9 | rs7780273 | chr7:7210818 | *C1GALT1* | intron | $3.60 \times 10^{-20}$ | $3.78 \times 10^{-07}$ |
| 10 | rs4559164 | chr7:128761756 | *CALU* | intron | $1.50 \times 10^{-02}$ | $1.79 \times 10^{-07}$ |
| 11 | rs339050 | chr7:128746151 | *CALU* | intron | $1.80 \times 10^{-02}$ | $3.41 \times 10^{-07}$ |
| 12 | rs10452178 | chr4:102892161 | *CISD2* | 3'UTR | $4.40 \times 10^{-06}$ | $4.50 \times 10^{-05}$ |
| 13 | rs223332 | chr4:102869029 | *CISD2* | 5'UTR | $1.30 \times 10^{-10}$ | $6.84 \times 10^{-07}$ |
| 14 | rs277646 | chr3:100354982 | *NIT2* | intron | $5.50 \times 10^{-09}$ | $1.46 \times 10^{-04}$ |
| 15 | rs1214375 | chr3:100346788 | *NIT2* | intron | $4.30 \times 10^{-09}$ | $5.96 \times 10^{-14}$ |
| 16 | rs11714045 | chr3:100336281 | *NIT2* | intron | $1.80 \times 10^{-06}$ | $1.58 \times 10^{-20}$ |
| 17 | rs3763196 | chr6:110999834 | *RPF2* | intron | $1.20 \times 10^{-14}$ | $4.77 \times 10^{-13}$ |
| 18 | rs9374241 | chr6:111006140 | *RPF2* | intron | $2.15 \times 10^{-07}$ | $2.20 \times 10^{-07}$ |
| 19 | rs6926809 | chr6:110984652 | *RPF2* | intron | $8.30 \times 10^{-07}$ | $2.39 \times 10^{-31}$ |
| 20 | rs9384778 | chr6:111002085 | *RPF2* | intron | $5.90 \times 10^{-13}$ | $5.13 \times 10^{-12}$ |
| 21 | rs9386979 | chr6:111002769 | *RPF2* | intron | $3.50 \times 10^{-08}$ | $2.94 \times 10^{-07}$ |
| 22 | rs9384779 | chr6:111003447 | *RPF2* | intron | $5.80 \times 10^{-15}$ | $9.62 \times 10^{-15}$ |
| 23 | rs75043369 | chr6:111019754 | *RPF2* | intron | $4.10 \times 10^{-08}$ | $7.02 \times 10^{-07}$ |
| 24 | rs56820058 | chr6:111014180 | *RPF2* | intron | $2.40 \times 10^{-08}$ | $6.87 \times 10^{-06}$ |
| 25 | rs9374247 | chr6:111020258 | *RPF2* | intron | $1.90 \times 10^{-09}$ | $1.27 \times 10^{-12}$ |
| 26 | rs1044388 | chr6:111025605 | *RPF2* | 3'UTR | $7.60 \times 10^{-14}$ | $1.20 \times 10^{-72}$ |
| 27 | rs10434875 | chr6:111028248 | *RPF2* | 3'UTR | $6.60 \times 10^{-08}$ | $5.17 \times 10^{-07}$ |
| 28 | rs5771217 | chr22:50189275 | *TRABD* | intron | $2.10 \times 10^{-06}$ | $5.76 \times 10^{-06}$ |

**Table 2 | The associations between identified apaQTL/eQTL-SNPs and LUAD risk**

| SNP | Gene | Location (GRCh38) | Alleles | MAF | OR (95% CI) [a] | *P* [a] |
|-----|------|-------------------|---------|-----|-----------------|---------|
| rs10452178 | *CISD2* | chr4:102892161 | C > T | 0.151 | 0.92 (0.87–0.98) | 0.009 |
| rs277646 | *NIT2* | chr3:100354982 | G > T | 0.058 | 1.12 (1.02–1.22) | 0.015 |
| rs11714045 | *NIT2* | chr3:100336281 | C > T | 0.226 | 1.05 (1.00–1.10) | 0.076 |

[a] Logistic regression analysis adjusted for age, gender, smoking pack-years (smoking status if pack-years information was not available in specific studies) and the top 10 Principal components.

was a significantly higher cell proliferation ability in the *NIT2*-rs277646-T group (*P* = 0.0027) (Fig. 4d–e).

To determine the effect of different alleles of rs277646 on tumor growth in vivo, SPCA1 cells transfected with *NIT2*-rs277646-T and *NIT2*-rs277646-G were injected into BALB/c-nu mice to construct an animal xenograft model (Fig. 4f). The results showed that the tumor volume growth rate in the *NIT2*-rs277646-G group was slower, while the tumor volume growth rate in the *NIT2*-rs277646-T group was faster and suddenly increased from the 22nd day (Fig. 4g, h). In addition, after killing the mice on the 30th day, the tumor weight of the *NIT2*-rs277646-T group significantly increased compared to the *NIT2*-rs277646-G group (Fig. 4i). Flow cytometry was performed on tumor tissue samples from a mouse model to detect and evaluate the apoptosis rate of the SPCA1 cell line under T and G alleles of rs277646 (Fig. 5a). The results showed that the apoptosis rate of the *NIT2*-rs277646-T group was significantly lower than that of the *NIT2*-rs277646-G group (Fig. 5b, c).

Further analysis of mouse tumor tissue sections using immunohistochemistry staining with cleaved-caspase 3 and ki67 showed that the positive rate of caspase-3 staining in the rs277646-T group was significantly lower than that in the rs277646-G group, while the positive rate of ki67 staining in the rs277646-T group was significantly higher than that in the rs277646-G group (Fig. 5d–g), indicating the T allele of rs277646 may promote the malignant phenotype of LUAD.

**Analysis of poly(A) sites of NIT2 based on the 3′RACE experiment**
According to the UCSC website (https://genome.ucsc.edu/), rs277646 is located at the 194 bp upstream of the 3′UTR of *NIT2* (Fig. 6a). Subsequently, through the NCBI website (https://www.ncbi.nlm.nih.gov/), it can be observed that the length of *NIT2* 3′UTR is 6367 bp. At the same time, there are 5 predicted poly (A) sites (PAS1-5) located at 949 bp, 961 bp, 1203 bp, 2472 bp, and 7233 bp, respectively. Subsequently, the 3′RACE experiment

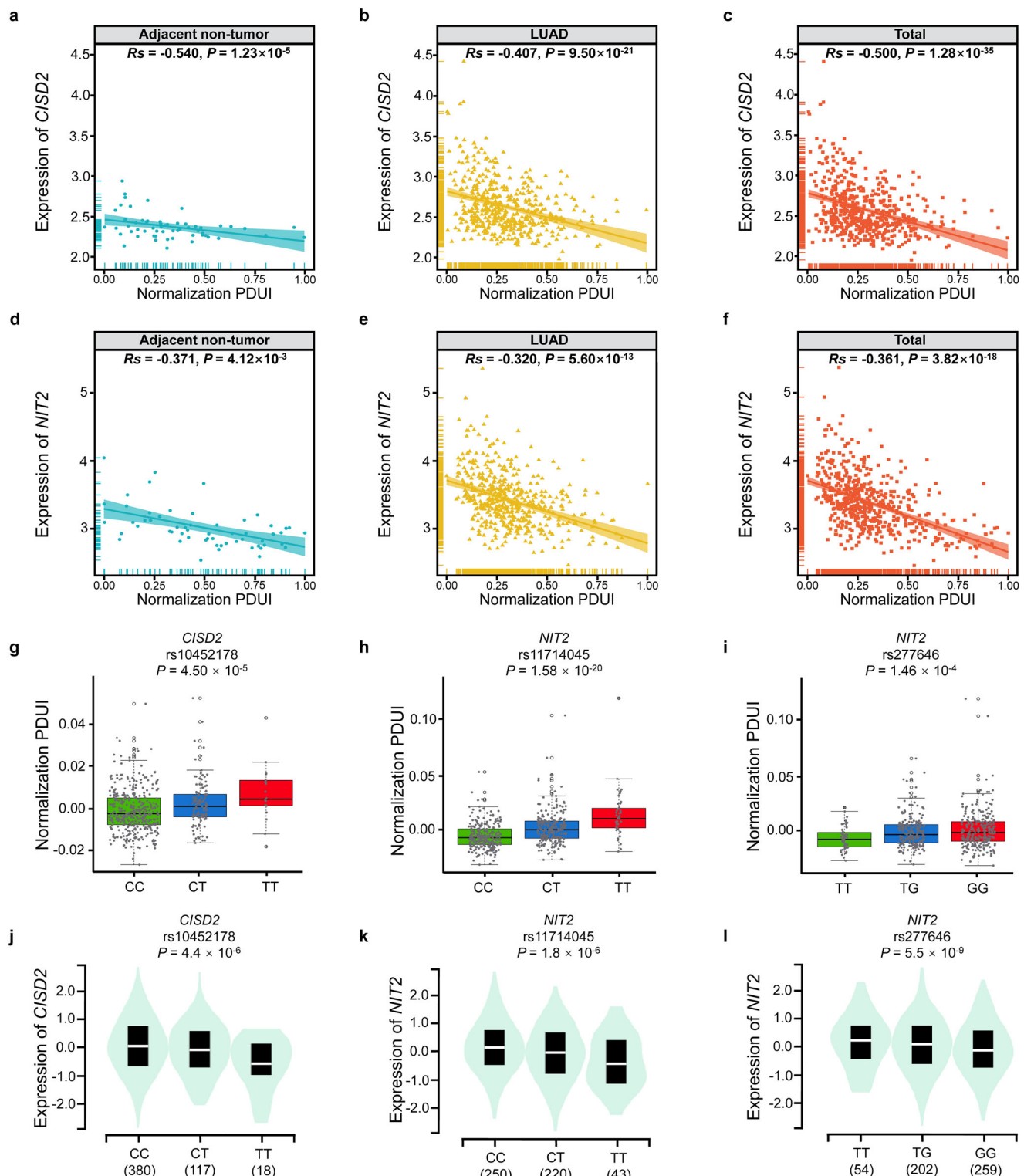

**Fig. 2 | Correlation between PDUI values and gene expression, apaQTL and eQTL analyses of 3 apaQTL/eQTL-SNPs. a–c** The correlation between PDUI value and *CISD2* expression in adjacent non-tumor tissues (**a**), LUAD tumor tissues (**b**) and total tissues (**c**). **d–f** The correlation between PDUI value and *NIT2* expression in adjacent non-tumor tissues (**d**), LUAD tumor tissues (**e**) and total tissues (**f**). **g–i** The relationship between different genotypes of rs10452178 and *CISD2* PDUI values (**g**), rs11714045 and *NIT2* PDUI values (**h**), rs277646 and *NIT2* PDUI values(**i**). **j–l** The relationship between different genotypes of rs10452178 and *CISD2* expression (**j**), rs11714045 and *NIT2* expression (**k**), rs277646 and *NIT2* expression (**l**).

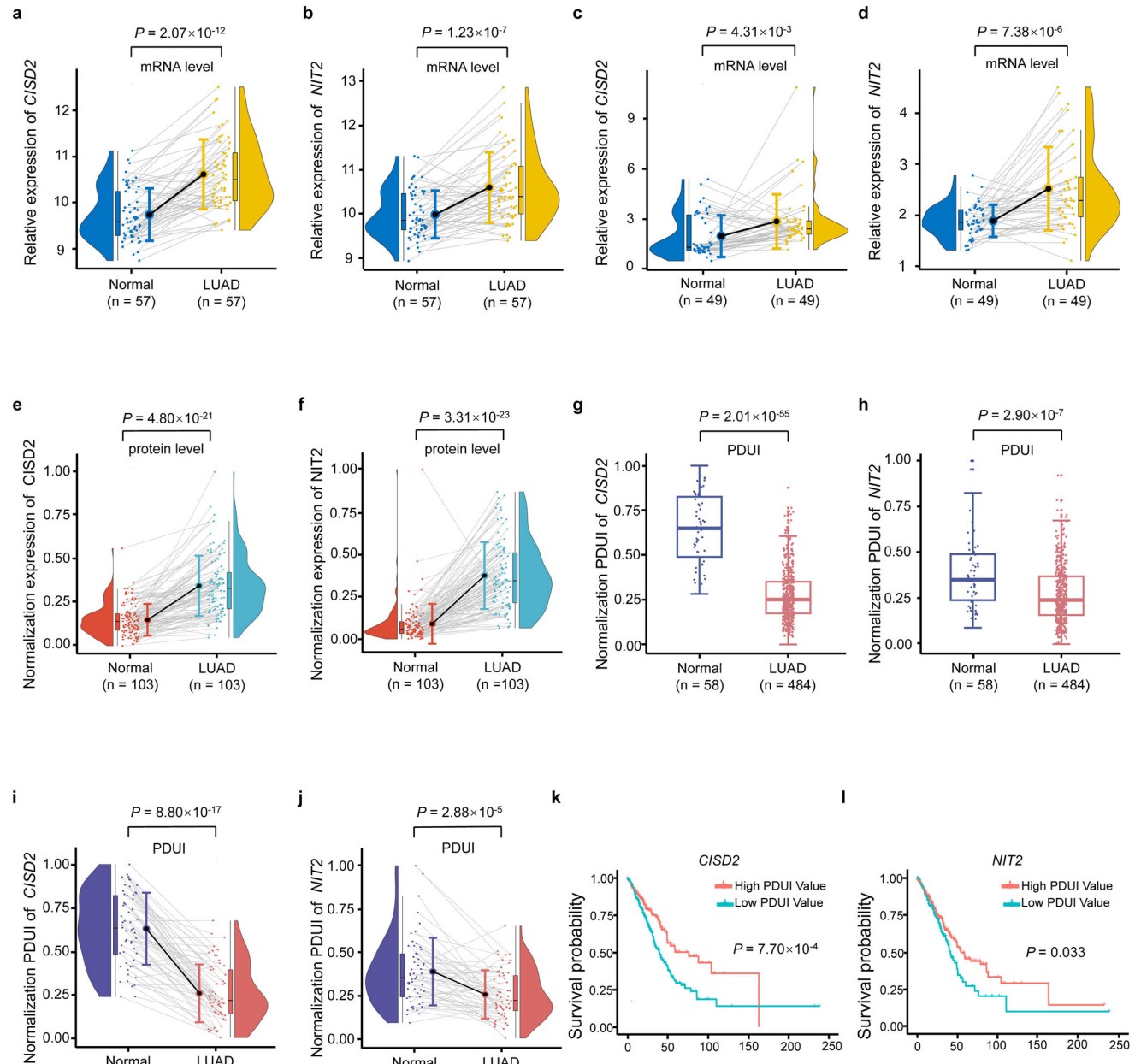

**Fig. 3 | Expression analysis and PDUI analysis of *NIT2* and *CISD2* between LUAD tumor tissues and adjacent non-tumor tissues, and survival analysis of *CISD2* and *NIT2*. a, b** *CISD2* mRNA expression (**a**) and *NIT2* mRNA expression (**b**) in the LUAD tumor tissues and paired adjacent non-tumor tissues in the TCGA dataset. **c, d** *CISD2* mRNA expression (**c**) and *NIT2* mRNA expression (**d**) in the LUAD tumor tissues and paired adjacent non-tumor tissues in the Chinese population. **e, f** CISD2 protein expression (**e**) and NIT2 protein expression (**f**) in the LUAD tumor tissues and paired adjacent non-tumor tissues in the Chinese population. **g, h** The PDUI value of *CISD2* (**g**) and NIT2 (**h**) between LUAD tumor tissues and adjacent non-tumor tissues. **i, j** The PDUI value of *CISD2* (**i**) and *NIT2* (**j**) between LUAD tumor tissues and paired adjacent non-tumor tissues. **k, l** Survival analysis of PDUI value of *CISD2* (**k**) and *NIT2* (**l**).

showed that the *NIT2* in SPCA1 cell line mainly selectively recognizes the PAS1 and PAS3 sites (Fig. 6b, c).

### The expression of NIT2 isoforms under different alleles of apaQTL/eQTL-SNP rs277646

The expression of the longer 3′UTR transcript of *NIT2* was significantly lower in the rs277646-T allele than that in the G allele ($P = 2.39 \times 10^{-4}$) (Fig. 6d), while the expression of the shorter 3′UTR transcript of *NIT2* was significantly higher in the rs277646-T allele than that in the G allele ($P = 1.72 \times 10^{-4}$) (Fig. 6e). Meanwhile, whether under the G allele or T allele of rs277646, the expression of the shorter 3′UTR transcript of *NIT2* was significantly higher than that of the longer 3′UTR transcript ($P = 7.46 \times 10^{-6}$,

$P = 3.58 \times 10^{-5}$) (Fig. 6f, g). At the same time, the ratio of the shorter 3′UTR transcript to the longer 3′UTR transcript under the T allele of rs277646 (ratio=11.19) was higher than that under the G allele of rs277646 (ratio=3.54). This indicates that the *NIT2* inclined to produce shorter 3′UTR transcripts under the T allele of rs277646.

### The impact of hsa-miR-650 binding to the long 3′UTR of *NIT2*

Due to the different genotypes of rs277646 affecting the expression of different subtypes of *NIT2*, we further investigated its influence on gene-miRNA interactions. Firstly, we selected microRNAs using the ENCORI database (http://starbase.sysu.edu.cn/) and miRDB (https://mirdb.org/ontology.html), ultimately identifying hsa-miR-650 and hsa-miR-642a-3p

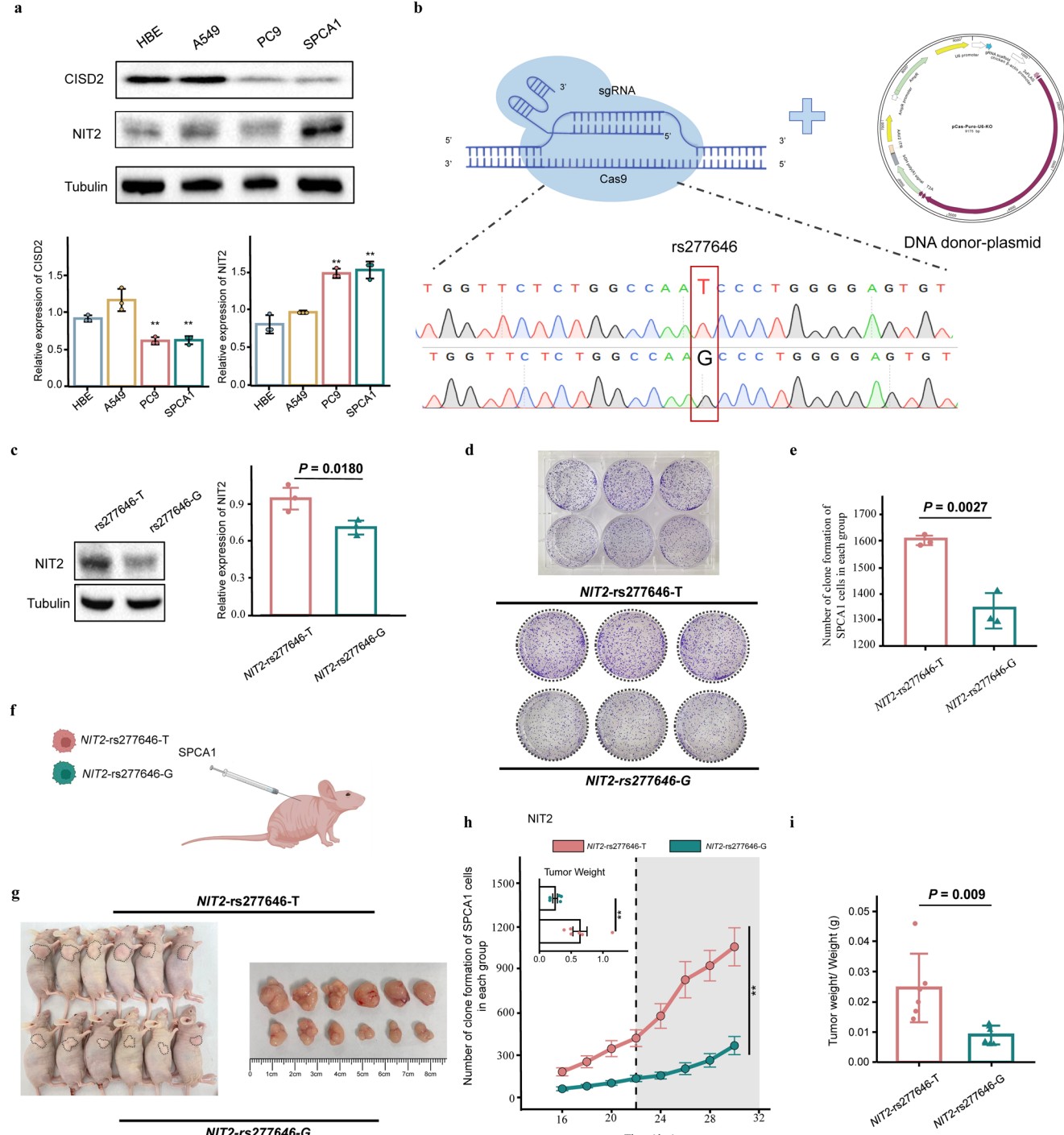

**Fig. 4 | The T allele of rs277646 is more likely to promote the proliferation in vitro and tumor growth in vivo. a** The expression of CISD2 and NIT2 in cell lines. **b** *NIT2*-rs277646-G was obtained through genome editing mediated by CRISPR/Cas9. **c** The expression of NIT2 in *NIT2*-rs277646-T and *NIT2*-rs277646-G. **d** SPCA1 cell proliferation experiment after transfection. **e** The rs277646-T can promote the proliferation of SPCA1 cells. **f** Xenotransplantation animal model. **g** Macroscopic observation of tumor nudity. **h** Comparison of tumor volume changes and weight between *NIT2*-rs277646-T group (*n* = 6 mice) and *NIT2*-rs277646-G group (*n* = 6 mice) in mice. **i** The ratio of tumor weight to body weight in mice between the *NIT2*-rs277646-T group and the *NIT2*-rs277646-G group *$P < 0.05$, * * $P < 0.01$, * * * $P < 0.001$.

as two microRNAs. According to the prediction results from the TargetScan website (https://www.targetscan.org/vert_80/), both microRNAs are predicted to bind between Poly(A)1 and Poly(A)3. Specifically, Poly(A)1 is located 83 bp downstream from the start site of the *NIT2* 3′UTR, while Poly(A)3 is situated 337 bp downstream from the start site of the *NIT2* 3′ UTR. hsa-miR-650 binds to the region spanning 248-255 bp of the *NIT2* 3′ UTR, whereas hsa-miR-642a-3p binds to the region spanning 320-326 bp of

the *NIT2* 3′UTR (Fig. 6h). To validate the involvement of microRNAs in this interaction, we constructed short sequences (~ PAS1) and long sequences (~ PAS3) for the luciferase reporter gene assay.

Results from the luciferase reporter gene assay showed that the hsa-miR-650 mimics or hsa-miR-642a-3p mimics did not bind to the short 3'UTR of the *NIT2* (~ PAS1). However, in the long 3'UTR of *NIT2* (~ PAS3), the has-miR-650 mimics significantly reduced luciferase

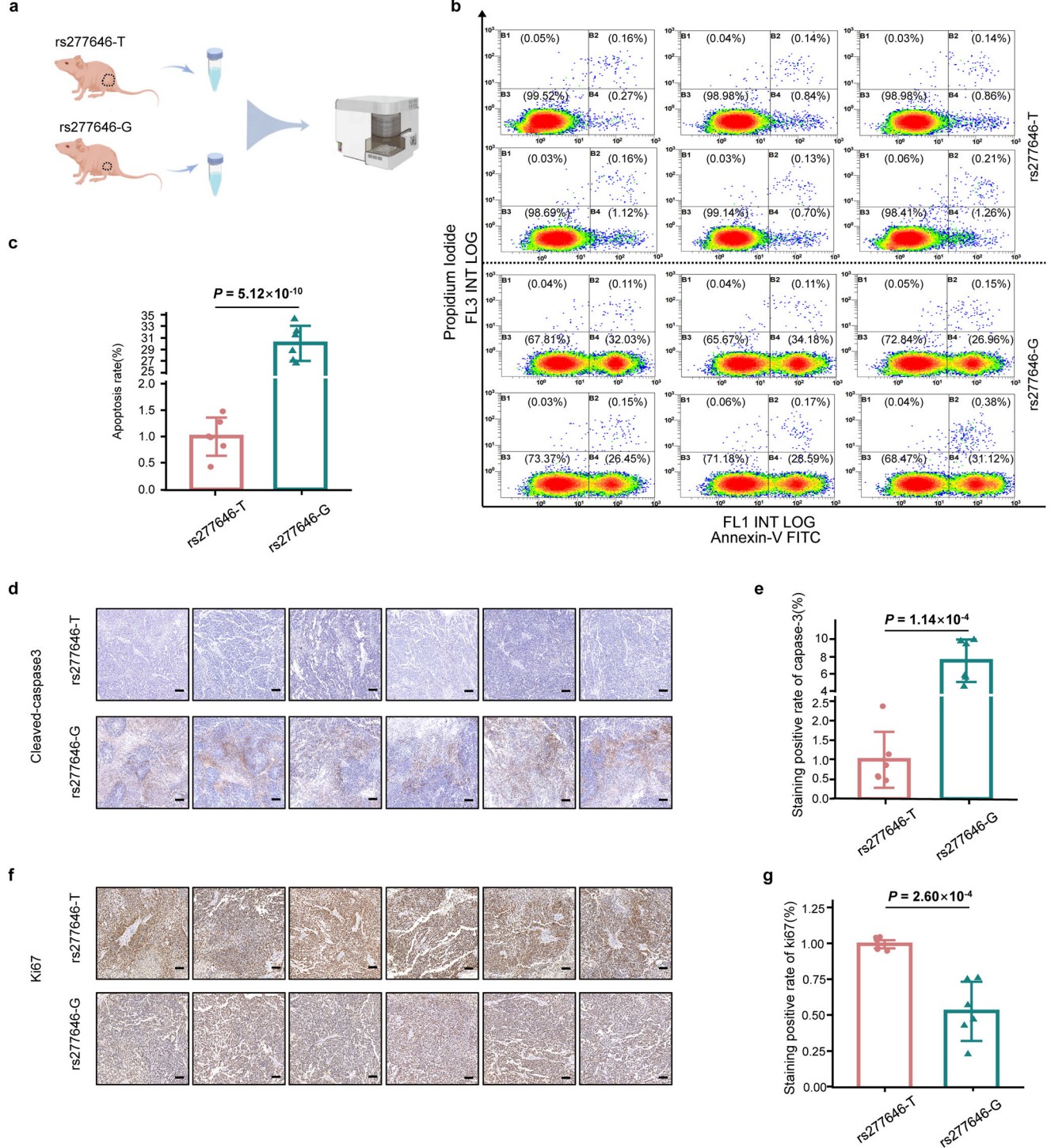

**Fig. 5 | The T allele of rs277646 is more likely to inhibit apoptosis and promote proliferation in the mice model. a** Flow cytometry pattern diagram of primary tumor cells extracted from mice. **b**, **c** Flow cytometry was used to detect apoptosis of annexin V/PI. Harvest cells and use annexin V and 1 μ PI staining. Flow cytometry showed that the cells were in the stages of living cells, early cell apoptosis, and late cell apoptosis. Compared with the rs277646-G group, the apoptosis rate of tumor cells under rs277646-T was reduced. **d** Image of tumor slices from xenograft animals stained with Cleared Caspase-3 (100x objective). **e** The positive rate of Cleared Caspase-3 in the rs277646-T group was significantly lower than that in the rs277646-G group. **f** Image of tumor slices from xenograft animals stained with ki67 (100x objective). **g** The positive rate of ki67 staining in the rs277646-T group was significantly higher than that in the rs277646-G group.

activity, indicating that the has-miR-650 mimics binds to the long 3'UTR of *NIT2* (~ PAS3); The luciferase activity of hsa-miR-642a-3p mimics did not decrease, indicating that hsa-miR-642a-3p mimics does not bind to the long 3'UTR of *NIT2* (~ PAS3) (Fig. 6i). Therefore, the rs277646-T genotype leads to *NIT2* preferentially utilizing the proximal poly(A) site, resulting in shorter 3'UTR transcripts, which leads to the loss of hsa-miR-

650 binding sites on *NIT2*, thereby affecting the expression levels of *NIT2*.

## Discussion

By integrating APA-related LUAD genes, 3'aQTL-atlas and eQTL analysis, we identified 28 candidate LUAD-related apaQTL/eQTL-SNPs, while

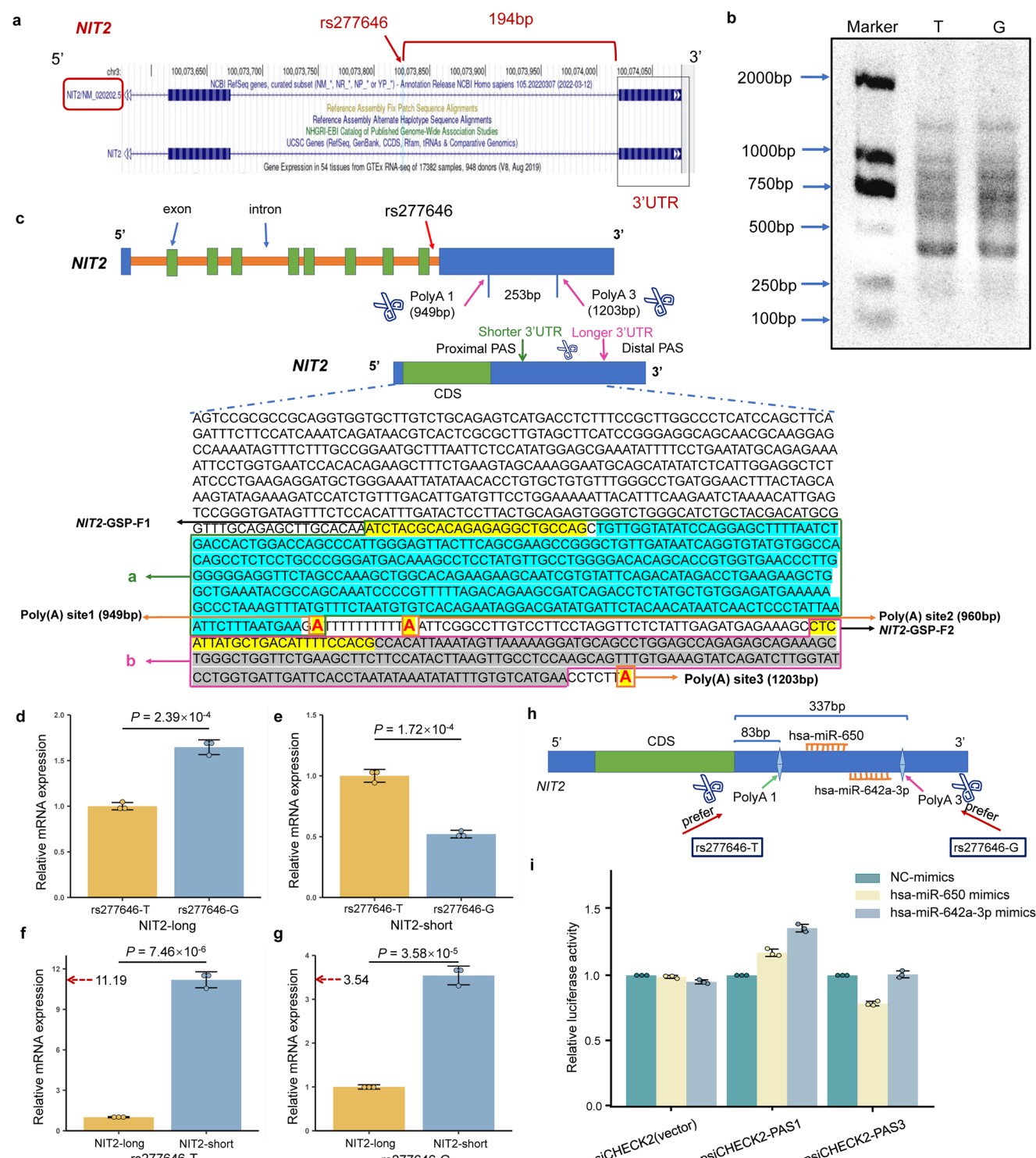

**Fig. 6 | Differential effects of rs277646 alleles regulating NIT2 3′UTR length and expression patterns. a** The relationship between rs277646 on the NCBI website and the 3′UTR of the target gene *NIT2*. **b** Agarose gel map of PCR products of 3′RACE under different alleles of rs277646. **c** Different alleles of rs277646 may mediate differences in poly (A) selection, resulting in different 3′UTR transcript schematics and PCR product sequences. **d** The expression of the longer 3′UTR transcript of *NIT2* in the rs277646-G allele is significantly higher than that in the T allele. **e** The expression of the shorter 3′UTR transcript of *NIT2* in the rs277646-T allele is significantly higher than that in the G allele. **f** In the sample of rs277646-T allele, the expression of the shorter 3′UTR transcript of *NIT2* was significantly higher than that of the longer 3′UTR transcript. **g** In the sample of rs277646-G allele, the expression of the shorter 3′UTR transcript of *NIT2* was significantly higher than that of the longer 3′UTR transcript. **h** Pattern Diagram of miRNA Binding to the NIT2 3′UTR Region. **i** Dual luciferase reporter gene assay for 3′UTR of *NIT2* and miR-650, miR-642a-3p; *n* = 3. 3′RACE, 3′ rapid amplification of cDNA ends; 3′UTR, 3′untranslated region; CDS, coding sequence; PAS, polyadenylation signal; GSP, gene-specific primer. *$P < 0.05$, **$P < 0.01$, ***$P < 0.001$.

rs277646, rs11714045, and rs10452178 were associated with the altered risk of developing LUAD based on the China Nanjing Lung Cancer GWAS database. Additionally, the rs277646 in *NIT2* may act as a causal variant associated with LUAD. The rs277646-T genotype causes *NIT2* to preferentially use the proximal poly (A) site, resulting in a shorter 3'UTR transcript and the loss of the hsa-miR-650 binding site, thereby affecting LUAD malignant phenotypes by regulating *NIT2* expression levels. Our findings may provide new insights into understanding and exploring APA events in LUAD carcinogenesis.

*NIT2*, identified as ω-amidase, plays an important metabolic role by catalyzing hydrolysis of α-ketoglutaramate and α-ketosuccinamate, yielding α-ketoglutarate and oxaloacetate, respectively[18,19]. Some studies have shown that the overexpression of *NIT2* is related to the occurrence and development of colon cancer[20,21] and tongue squamous cell carcinoma[22]. In colon cancer, the downregulation of *NIT2* inhibits the proliferation of colon cancer cells and induces cell cycle arrest through the caspase-3 and PARP pathways. Meanwhile, low expression of *NIT2* may not only inhibit the growth of colon cancer cells, but also promote apoptosis of cancer cells, indicating that *NIT2* may play a role in promoting cancer[20].

The rs277646 is located 194 bp upstream of the 3'UTR end of *NIT2*. In the APA analysis, the PDUI value of rs277646 decreased after mutation. According to the definition of PDUI value[23], which means that if the PDUI value decreases, *NIT2* may tend to use the proximal poly (A) site, resulting in producing transcripts with shorter 3'UTR. At present, eQTL analysis can provide clues for the correlation between SNPs and gene expression, while apaQTL analysis, as a bridge connecting functional SNPs and gene expression, effectively explains the intermediate molecular mediated mechanism of abnormal gene expression regulated by different SNPs genotypes. The eQTL analysis indicated rs277646 G > T increases the expression of *NIT2* in lung tissue, which was consistent with the results of APA analysis. At the same time, in the population of China Nanjing Lung Cancer GWAS database, we found that rs277646 G > T increased the risk of developing LUAD. It is biological plausible that rs277646 G > T affects the occurrence of APA events, making *NIT2* tend to use the proximal poly (a)

site, resulting in producing transcripts with shorter 3'UTR. This may make it easier to evade the negative regulation of microRNAs (miRNAs) or RNA binding proteins (RBPs), thereby enhancing its mRNA expression and protein translation. Resulting in an increase in the expression of *NIT2* in LUAD tumor tissues, this in turn may affect protein expression levels, population susceptibility, and disease outcomes (Fig. 7).

Combined with the ENCORI database (http://starbase.sysu.edu.cn/) of the bioinformatics prediction website, we explored the interaction between rs277646 mediated changes in the 3'UTR length in *NIT2* and miRNA and RBPs. The results indicate that there is a significant correlation between the expression of RBPs and *NIT2*, such as ALYREF, RBMX, EIF4A3, U2AF1, PCBP2, DHX36, IGF2BP3, etc. And the expression of these RBPs is positively correlated with the expression of *NIT2*. So, with the mutation of rs277646 G > T, the 3'UTR of *NIT2* is shortened, and the binding RBP sites are reduced, making RBP unable to bind, resulting in freer RBP and more expression of *NIT2*. Therefore, mutations in rs277646 G > T can lead to APA events, which may alter the length of 3'UTR by regulating the binding of RBPs to the target gene, thereby affecting the stability, expression level, and translation ability of the target gene[24].

This study has the following highlights. Firstly, we accurately mapped apaQTL/eQTL-SNPs from 3'aQTL-atlas to APA-related LUAD genes. Secondly, APA-related LUAD genes were identified not only through correlation analysis between PDUI value and gene expression, but also according to the consistently differential expressed genes both at the mRNA and protein levels. Thirdly, the screening of SNPs with both apaQTL and eQTL functions (apaQTL/eQTL-SNPs) may help well explain the intermediate mediation mechanism of eQTLs and complement the shortcomings of eQTL analysis. Finally, a large sample population (8762 LUAD cases and 13,328 cancer-free healthy controls) of the susceptibility study may make our conclusions more reliable and persuasive.

Although our research indicates a potential significant association between candidate SNPs and LUAD in the Chinese population, some limitations remain. Firstly, features evaluated in European or American populations may be irrelevant or insensitive to the effects of these SNPs.

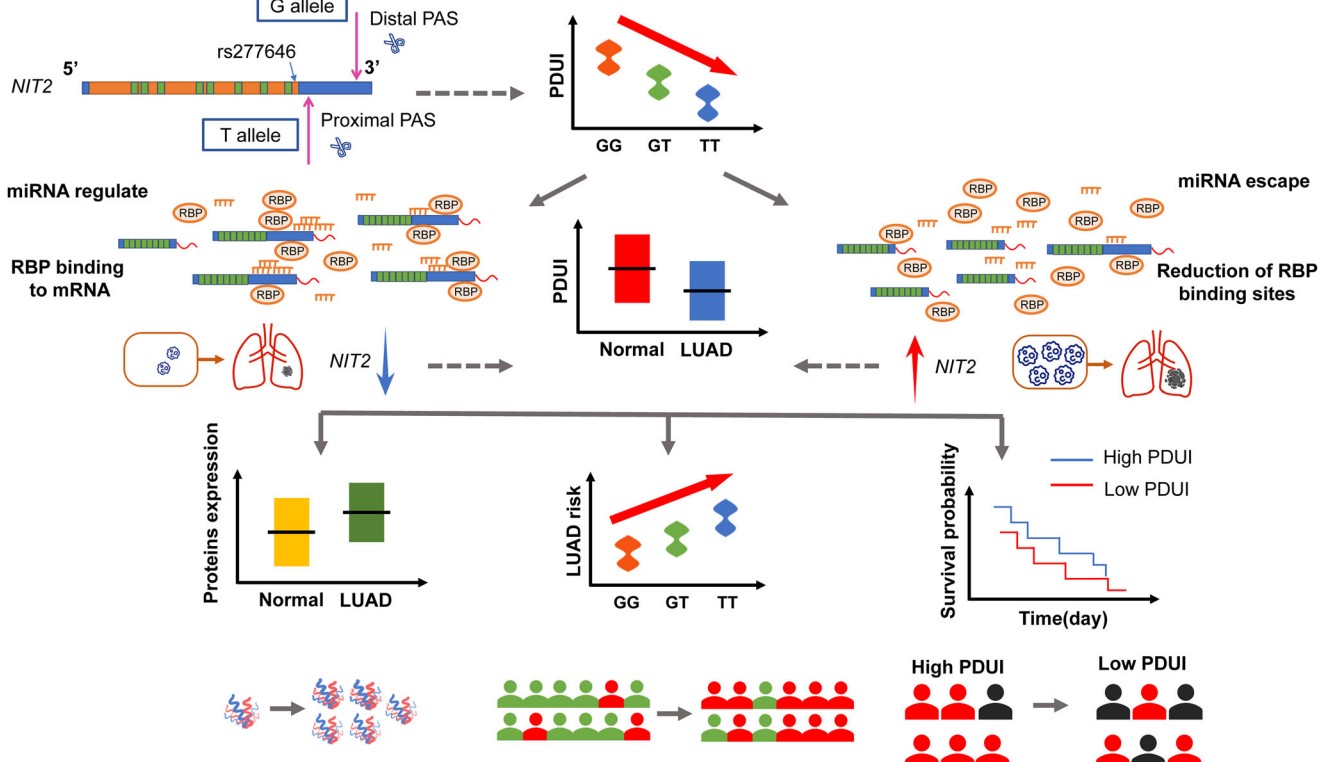

**Fig. 7 | The potential regulation mechanism of rs277646 mediating malignant phenotypic changes in LUAD by regulating the 3'UTR length of *NIT2*.**

Typically, larger sample sizes are required to achieve sufficient statistical power, especially when exploring genetic variations that may have moderate effects on phenotypes. Secondly, despite in vitro and in vivo experiments indicated the causal potential of rs277646 mediating malignant phenotypic changes in LUAD, however, we could not ignore the fact that the rs277646 in China Nanjing Lung Cancer GWAS have not reached genome-wide significance level, which needs further validation in larger sample of Chinese LUAD cases and controls.Thirdly, it's important to acknowledge the influence of genetic heterogeneity within and between populations. Despite the China Nanjing Lung Cancer GWAS database appearing to be predominantly composed of the Chinese Han population, the public 3′aQTL-atlas and eQTL summary is based on European population data from the GTEx consortium. Due to these population differences, MAF and LD may vary between the two ancestries, potentially affecting the results of GWAS studies. In subsequent research, consideration of the impact of racial differences may also be necessary. Fourthly, although we rigorously filtered based on consistent expression patterns between proteins and mRNA, when selecting APA-related genes, using $|Rs| > 0.3$ and $P_{FDR} < 0.05$ as criteria for defining genes related to APA events might be too loose, thus potentially compromising the accuracy and interpretation of the results. Finally, the correlation between PDUI and *NIT2* expression in LUAD tumor tissues is moderate, with a $|Rs|$ value of 0.320, slightly exceeding 0.3, suggesting a potential risk of insufficient correlation significance, which may warrant further consideration and validation. Therefore, this factor should be carefully considered when interpreting and inferring these results.

## Methods

### Selection of APA-related LUAD genes

In the global APA events of different cancer types from TCGA database characterized by Xiang et al., they systematically assessed the correlation between the percentage of distal poly(A) site usage index (PDUI) values and expression levels of the corresponding genes using Spearman's correlation ($Rs$). The genes related to the APA events were defined as $|Rs| > 0.3$ and a false discovery rate (FDR) of $P_{FDR} < 0.05$. Then, we obtained APA-related genes in LUAD from this published dataset[14].

Based on those APA-related genes in LUAD, we further identified significantly differential expressed genes (mRNA level) between 57 paired LUAD tumor tissue and adjacent non-tumor tissues through TCGA database by setting fold changes (FC) with $|FC| > 1.5$ and statistically significant with $P < 0.05$. We then validated those genes (mRNA level) using the comprehensive transcriptomics database of LUAD (49 paired LUAD tumor tissue and adjacent non-tumor tissues) from the Chinese population (GSE140343)[25], and obtained the validated differential expressed genes (mRNA level) in Chinese.

Considering the low consistency rate of differentially expressed mRNAs and its′ coding proteins[26], we then combined with the proteomic database of the above same Chinese population (IPX0001804000) to obtain LUAD related genes (protein level) that also differently expressed at their coding protein level. Finally, the overlapped APA-related LUAD genes with consistent differential expression at both the mRNA and its′ coding protein levels were obtained.

### Selection of apaQTL/eQTL-SNPs in APA-related LUAD genes

apaQTL-SNPs that locate on the above identified APA-related LUAD genes were downloaded from the 3′aQTL-atlas online website (https://wlcb.oit.uci.edu/3aQTLatlas). In 3′aQTL-atlas online website, Cui et al. used the Dapars algorithm to calculate PDUI values from RNA-seq data from GTEx database, thereby measuring the association between different genotypes of SNPs and PDUI values of target genes[17]. Then, apaQTL-SNPs were defined based on within a 1-MB interval of the target APA-related genes of the 3′ UTR regions and $P_{FDR} < 0.05$.

Based on those candidate apaQTL-SNPs, we further performed eQTL analysis ($P < 0.05$) from the GTEx database (https://gtexportal.org/home/), thus obtaining SNPs both with apaQTL and eQTL functions. Subsequently, candidate apaQTL/eQTL-SNPs with minor allele frequency (MAF) > 0.05

in Chinese Han population (CHB) were screened. Finally, we performed linkage disequilibrium (LD) analysis with an $r^2$ threshold of 0.80 to further screen the candidate apaQTL/eQTL-SNPs.

### Study population of the susceptibility study

Genetic susceptibility research is typically conducted in large-scale populations, employing genome-wide association studies (GWAS). These studies compare genetic variations (single nucleotide polymorphisms, SNPs) among individuals to explore their relationship with specific diseases or traits. In our study, a total of 8762 LUAD cases and 13,328 healthy controls were included in the susceptibility study. Details have been mentioned in the study published by Dai et al.[27]. All these LUAD cases were histologically confirmed by at least two pathologists who had not received chemotherapy or radiotherapy before diagnosis. A healthy control group was selected from participants in community screening of non-communicable diseases and frequency matched with cases by age, gender, and geographical regions. All participants signed informed consent before this study. The Committee of the Human Ethics Committee of Nantong University approved the study protocol. All ethical regulations relevant to human research participants were followed.

### Cell culture

The LUAD cell lines used in this study were A549, PC9 and SPCA1, as well as HBE (human bronchial epithelial cell line), all cell lines were purchased from American Type Culture Collection (ATCC). The cell lines were cultured in DMEM medium supplemented with 10% fetal bovine serum, 100 U/mL penicillin, and 100% μ G/mL streptomycin in an incubator with 5% $CO_2$ at 37 °C. HEK 293 T cells were cultured in Dulbecco′s modified Eagle′s medium (DMEM) supplemented with 10% FBS, 100 U/mL penicillin, and 100 μg/mL streptomycin at 37 °C with 5% $CO_2$.

### Western blot analysis

Western blot analysis was conducted and the antibodies we used were anit-NIT2 (1:5000, Proteintech), anti-CISD2 (1:2000, Proteintech). The protein level of NIT2 and CISD2 was standardized by Tubulin (1:1000, Beyotime). The antibodies used in this study are shown in Supplementary Table 1. Image J software (Version 1.54) was used for gray-scale analysis of protein bands.

### Cell transfection

The commercialized CRISPR/Cas9 expression vectors expressing Cas9, puromycin resistant *NIT2*-rs277646-G gene and sgRNA were designed and synthesized by Nanjing Corues Biotechnology Co., Ltd. According to the manufacturer's instructions, the single-stranded DNA donor and *NIT2* CRISPR/Cas9 plasmid were co-transfected into SPCA1 cells using Lipofectamine 3000 (Invitrogen, Grand Island, NY, USA). After 24 h, the medium was replaced with DMEM medium containing 2 μg/μl puromycin (VWR Pty Ltd, Brisbane, Australia), and cultured for 48 h. The single-cell-derived colonies were obtained by the surviving cells. Cells were then harvested for DNA sequencing to verify mutation efficiency. Oligo sequence is shown in Supplementary Table 2.

### Cell proliferation assay

A total of 1000 *NIT2*-rs277646-G and *NIT2*-rs277646-T SPCA1 cells were seeded into a 6-well plate and cultured in an incubator with 5% $CO_2$ at 37 °C. The DMEM medium were changed every 3 days. After two weeks, the cells were fixed and stained with 0.1% crystal violet (Beyotime, Shanghai, China) for 30 min. Image J software was used for cell counting.

### Flow cytometry for apoptosis analysis

Cells were harvested into 5 ml centrifuge tubes at 37 °C and washed with PBS. After adding 195 μL Annexin-V-FITC conjugate, 5 μL Annexin-V-FITC and 10 μL PI stain, the cells were incubated at room temperature for 15 min in the dark. The apoptotic cells were measured using the FACScan flow cytometer (Becton, CA) equipped with CellQuest Software (Becton

Dickinson). The gating strategies of flow cytometry in this study see supplementary Fig. 1.

## Tumor xenograft model

The animal experiments were approved by the Institutional Animal Care and Use Committee of Nantong University (Approval No: S20220224-006). BALB/c male mice (4-5 weeks old) were purchased from SLAC Experimental Animal Center (Shanghai, China) and maintained in SPF facilities. The mice were randomly divided into two groups ($n = 6$ for each group). $5 \times 10^6/100\ \mu L$ of NIT2-rs277646-T or NIT2-rs277646-G SPCA1 cells were subcutaneously injected into the right axilla of BALB/c mice. The tumor volume was measured using a vernier caliper at the specified time and the tumor volume using the formula: L (length)×W(width)$^2$×$2^{-1}$. At the end of the experiment, all mice were euthanized, and the tumors were photographed and weighed before freezing for further immunohistochemical analysis. According to the humane endpoint evaluation criteria established by the Institutional Animal Care and Use Committee (IACUC) of the Institute of Laboratory Animals, Nantong University, the tumor weight should not exceed 10% of the animal's body weight, which corresponds to a maximum subcutaneous tumor diameter of 20 mm in a 25 g mouse. In this study, the tumor burden in all animals did not exceed these limits. For immunohistochemical staining (IHC), we executed it according to standard protocols. The following antibodies were used for IHC: ki67 (GB111141-100, Servicebio, Shanghai, China) and cleaved caspase 3 (9664 S, CST, USA). We have complied with all relevant ethical regulations for animal use. The antibodies used in this study are shown in Supplementary Table 1.

## Rapid amplification of 3′- cDNA ends (3′RACE) experiment and qRT-PCR Assay

The 3′RACE experiment was used to identify different poly (A) sites of NIT2 gene with a 3′RACE kit (Jingrui, Guangzhou, China). The forward sequences of gene specific primers (GSP-F) for NIT2 are shown in Supplementary Table 3. The reversed sequences of GSP (GSP-R) were provided by the 3′RACE kit. The qualified products of 3′RACE were sequenced by Sanger sequencing.

To quantity the target long and short 3′UTR sequences of NIT2, qRT-PCR was performed and the relative quantitative value for NIT2 was determined as the $2^{-\triangle\triangle CT}$. GAPDH was used as an internal reference gene. The target long and short 3′UTR sequences of NIT2 and GAPDH sequences are shown in Supplementary Table 3.

## Fluorescent enzyme reporter gene assay

Fluorescent enzyme activity was measured using the Dual-Luciferase Reporter Assay System (Promega). According to the manufacturer's instructions, 293 T cells with a transfection efficiency of 40% were transfected with the transfection reagent Fitran (HuiJun, Guangzhou, China) in a 24-well plate. The psiCHECK2 vector was obtained from Jingrui (Guangzhou, China). Transfection groups were as follows: (i) Blank control (transfection reagent only), (ii) psiCHECK2(empty vector) + miRNA NC, (iii) psiCHECK2(empty vector) + hsa-miR-650 mimics, (iv) psiCHECK2(empty vector) + hsa-miR-642a-3p mimics, (v) psiCHECK2-PAS1 + miRNA NC, (vi) psiCHECK2-PAS1 + hsa-miR-650 mimics, (vii) psiCHECK2-PAS1 + hsa-miR-642a-3p mimics, (viii) psiCHECK2-PAS3 + miRNA NC, (ix) psiCHECK2-PAS3 + hsa-miR-650 mimics, (x) psiCHECK2-PAS3 + hsa-miR-642a-3p mimics. Cells were collected 48 h post-transfection, and the Dual-Luciferase Reporter Assay System (Promega, E1910) was used for analysis.

## Statistics and reproducibility

The expression differences of mRNA or protein, as well as the differences of PDUI values between LUAD tumor tissue and adjacent non-tumor tissues were analyzed by independent Student's t test or paired t test, the centers and the error bars represent the mean and the SD, respectively. Logistic regression analyses were applied to evaluate the associations between apaQTL/eQTL-SNPs and the risk of developing LUAD based on the odds ratios (ORs) and 95% confidence intervals (CIs), adjusting for age, gender, smoking pack-years (smoking status if pack-years information was not available in specific studies) and the top 10 Principal components. To avoid potential batch effects from gene expression quantification, gene expression was recalculated as reads per kilobase per million mapped reads (RPKM) for all TCGA samples when calculating the correlation between gene expression and PDUI value. Spearman correlation analysis was used to analyze and evaluate the correlation between PDUI values and corresponding gene expression. Violin plots of CISD2 and NIT2 expression for different genotypes in lung tissue (Fig. 2j–i) were obtained from the GTEx Portal (https://gtexportal.org/home/testyourown). Kaplan Meier survival analysis was used to evaluate the effect of PDUI values of target genes on the overall survival of LUAD patients. Due to the different expression ranges of genes in various databases, normalization of the data was necessary for reliable and meaningful comparisons. We balanced these values through a min-max normalization process, which transformed them into a range of (0,1). Log-rank test was used to compare the difference of survival rate between different groups. All bar graphs are representative of three or more independent experiments as indicated in the figure legends. All statistical tests were two-sided, and P-values < 0.05 were considered statistically significant. All statistical analyses were performed by R version 4.1.1 software.

## Reporting summary

Further information on research design is available in the Nature Portfolio Reporting Summary linked to this article.

## Data availability

The gene expression data of 57 paired LUAD tumor tissues and adjacent non-tumor tissues were sourced from the publicly available TCGA database (TCGA-LUAD) (https://portal.gdc.cancer.gov/projects/TCGA-LUAD). The comprehensive transcriptomics database of LUAD (49 paired LUAD tumor tissue and adjacent non-tumor tissues) from the Chinese population are available at the Gene Expression Omnibus (GEO) repository under Study Accession GSE140343, and the proteomic database of the above same Chinese population are available at iProx Consortium with the subproject ID IPX0001804000. The data on PDUI values in APA events are available in the Synapse database https://www.synapse.org, login number is syn7888354. The China Nanjing Lung Cancer GWAS data is not publicly accessible, but any organization can apply to this team (M.Z. and H.S.) for this GWAS data. All other data are available from the corresponding author (or other sources, as applicable) on reasonable request. The numerical source data behind the graphs in the manuscript can be found in supplementary data. The uncropped gels are shown in supplementary Figs. 2–3.

## Code availability

The code used in this study can be found in the supplementary documents.

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

## Acknowledgements
The authors wish to thank all the study participants, research staff and students who participated in this work. This work was supported by the National Natural Science Foundation of China (82273715, 82203771), the National Key Research and Development Program of China (2022YFC2503202), the Science and Technology Program of Nantong City (MS22022062, JC22022002, JC22022004), and the Postgraduate Research and Practice Innovation Program of Jiangsu Province (KYCX23-3436).

## Author contributions
Conceptualization, Huiwen Xu, Yutong Wu, Qiong Chen and Yuhui Yu; Data curation, Na Qin and Wendi Zhang; Formal analysis, Huiwen Xu; Funding acquisition, Minjie Chu; Methodology, Tian Tian, Jiahua Cui and Lei Zhang; Software, Qianyao Meng, Xiaobo Tao and Qiong Chen; Supervision, Minjie Chu, Jiahua Cui and Tian Tian; Validation, Na Qin, Hongxia Ma and Minjie Chu; Visualization, Huiwen Xu, Yutong Wu and Siqi Li; Writing – original draft, Huiwen Xu, Yutong Wu, and Minjie Chu; Writing – review & editing, Huiwen Xu, Yutong Wu, Qiong Chen, Yuhui Yu, Qianyao Meng, Na Qin, Wendi Zhang, Xiaobo Tao, Siqi Li, Tian Tian, Lei Zhang, Hongxia Ma, Jiahua Cui, Minjie Chu. All authors have read and agreed to the published version of the manuscript.

## Competing interests
The authors declare no competing interests.

## Informed Consent Statement
Informed consent was obtained from all subjects involved in the study.
