## [Peer review file · Communications Biology]

Reviewers' comments:

Reviewer #1 (Remarks to the Author):

In the reviewed manuscript, Xu et al. identified SNPs with both eQTL and apaQTL functions based on the existing databases. Then, they performed functional experiments to validate the effects of the identified apaQTL/eQTL-SNP on the malignant LUAD in vitro and in vivo. While this study offers an interesting approach to interpreting cancer non-coding risk variants, I have concerns about the multiple data from inconsistent ancestries. Many other concerns still need to be addressed before further considering the manuscript.

Major:

1. "The variant T allele of rs10452178 located in CISD2 was significantly associated with a decreased risk of LUAD (OR= 0.92, P = 0.009), while the variant T allele of rs277646 located in NIT2 was significantly associated with an increased risk of LUAD (OR= 1.12, P = 0.015)." The GWAS information is from the Nanjing Lung Cancer GWAS database. Is this database publicly accessible? Additionally, it is unclear whether the p-value reaches genome-wide significance in the GWAS study, which may result in false-positive. Did the authors adjust the p-value? Also can these SNPs be identified in large-scale European-based GWAS studies?

2. The Nanjing Lung Cancer GWAS database seems to mainly consist of the Chinese Han population. Whereas the public 3'aQTL-atlas and eQTL summary is based on European population data from the GTEx consortium. Due to this difference in population, the minor allele frequency and LD may vary between the two ancestries.

3. In "Table 1 Detail information of the 28 candidate apaQTL/eQTL-SNPs", it seems that there are multiple SNPs associated with each gene. However, is it possible to determine whether their effect sizes are in the same direction? Furthermore, if each SNP has a significant, positive or negative effect on a gene's molecular phenotype (such as expression or APA), how can we identify the causal SNP?

4. The finding of the intronic variant rs277646 effect on 3'UTR length is interesting, but the explanation of the mechanism is quite descriptive and superficial. The authors need to conduct experiments to confirm which RBP/microRNA was indeed involved in this interaction. Without this evidence, it is difficult to evaluate the significance of this finding.

Minor:

1. lack reference.

-Line 56," Lung adenocarcinoma (LUAD) and lung squamous cell carcinoma (LUSC) have shown distinct incidence trends in recent decades..." lack reference.

-Lines 61-63 lack reference.

2. The expression of NIT2 in LUAD cell lines A549 was not significantly higher than that of the control in Figure 4A.

3. The method section "Study population of the susceptibility study" is confusing; what is a susceptibility study? There's not much mentioned in the main text about the results of this section.

4. Typo error:

-Spearman's rank correlation coefficient is often denoted by the Greek letter ρ (rho) or r_s (italic, please).

-Gene names should be in italics.

-3'UTR, 3' RACE.

Reviewer #2 (Remarks to the Author):

Xu et al aimed to investigate the association between the risk of LUAD and apaQTL/eQTL-SNPs. APA-related LUAD genes were screened, and candidate SNPs with both apaQTL and eQTL functions were identified. Large-scale GWAS analysis confirmed significant associations between rs10452178, rs277646 alleles, and LUAD risk. Functional experiments validated that the rs277646 variant T allele was associated with increased NIT2 expression, shorter 3'UTR transcript length, and promoted malignant phenotypes and tumor growth in LUAD. These findings reveal the potential impact of apaQTL/eQTL-SNPs on LUAD and provide new insights into the pathogenesis of LUAD. The discoveries in this study may contribute to the early risk assessment and treatment strategies for LUAD.

1. The design was not perfect when selecting APA related genes. The criteria of $|R_s| > 0.3$ and $FDR < 0.05$ were used in the selection of APA-related genes. These thresholds may be too loose in the definition of genes associated with APA events, which may result in too many genes being included in the study, potentially affecting the accuracy and interpretation of the results.

2. In this study, a particular concern was whether the correlation coefficients observed in cancer samples (line 315) reached a threshold that is generally considered significant in scientific studies. In the study, a correlation coefficient of about 0.3 is considered a low correlation level. Therefore, if the correlation coefficient in the cancer sample is lower than this value, there may be a risk that the association is not significant enough, which may require

further consideration and validation. In scientific research, it is essential to ensure the rigor of data analysis and the reliability of results. Therefore, this factor should be carefully considered when interpreting and inferring these results.

Reviewer #3 (Remarks to the Author):

Xu et al. integrated APA-related LUAD genes according to the consistently

differential expression both at the mRNA and protein levels, and then mapped apaQTL/eQTL-SNPs from 3'aQTL-atlas to APA-related LUAD genes. In the following susceptibility study, they evaluated the association between selected apaQTL/eQTL-SNPs and LUAD risk through 8,762 LUAD cases and 13,328 cancer-free healthy controls. The authors conclude that the variant T allele of rs277646 is associated with shortened 3'UTR length of NIT2 mRNA and may increase the risk of developing LUAD under the evidence that the variant T allele of rs277646 promoted malignant phenotypes in LUAD cell line and tumor growth in mice. Overall, this study is innovatively designed and carefully carried out, which may provide novel clues for understanding and exploring APA events in LUAD carcinogenesis. The manuscript would benefit from some additional information and please see comments below:

- 1) Legends should be short stand-alone described that allows the readers to understand the Figures.
- 2) In Figure 2, how the PDUI values were normalized?
- 3) There is nothing called borderline significant. It should be revised.
- 4) It is valuable that the authors considered the low consistency rate of differentially expressed mRNAs and its' coding proteins, what is the consistency rate in this study? Are the results similar to other studies?
- 5) In Figure 4A. The results showed that compared with HBE cell lines, NIT2 was overexpressed in the PC9 and SPCA1 cells. Why only SPCA1 was chosen to construct the mutant cell instead of the PC9?
- 6) In the study, the authors suggested that the NIT2 rs277646 variant T allele is associated with shortened 3'UTR length of NIT2 mRNA and may increase its expression hence may affect malignant phenotypes in LUAD cell line and tumor growth in mice. In Figure 4B, the authors successfully obtained NIT2-rs277646-G through CRISPR/Cas9 mediated genome editing. Does the mutation of NIT2-rs277646 from T to G affect NIT2 expression? These data are missing.

Dear Editors and Reviewers,

Thanks a lot for your and the reviewers' good comments about our manuscript (the manuscript number: COMMSBIO-24-0936, we have revised our manuscript very carefully according to your comments. The changes have been highlighted using a yellow font in the revised manuscript and the point-by-point responses were given as follows.

Reviewer #1

In the reviewed manuscript, Xu et al. identified SNPs with both eQTL and apaQTL functions based on the existing databases. Then, they performed functional experiments to validate the effects of the identified apaQTL/eQTL-SNP on the malignant LUAD in vitro and in vivo. While this study offers an interesting approach to interpreting cancer non-coding risk variants, I have concerns about the multiple data from inconsistent ancestries. Many other concerns still need to be addressed before further considering the manuscript.

Major:

1. "The variant T allele of rs10452178 located in *CISD2* was significantly associated with a decreased risk of LUAD (OR= 0.92, $P = 0.009$), while the variant T allele of rs277646 located in *NIT2* was significantly associated with an increased risk of LUAD (OR= 1.12, $P = 0.015$)." The GWAS information is from the Nanjing Lung Cancer GWAS database. Is this database publicly accessible? Additionally, it is unclear whether the p -value reaches genome-wide significance in the GWAS study, which may result in false-positive. Did the authors adjust the p -value? Also can these SNPs be identified in large-scale European-based GWAS studies?

Response: Thank you for your interest in our research and for raising these important questions. The GWAS information is sourced from the China Nanjing Lung Cancer GWAS database. The database originates from the study by Dai et al. ¹, which includes 13,327 lung cancer cases and 13,328 controls from the Chinese population

(Table S1).

Table S1. Sample Population Data Table

Characteristic	Lung cancer cases	Control
GSA array	10248	9298
NJMU GWAS	2126	3077
NJMU OncoArray	953	953
Total	13327	13328

The GWAS summary statistics of the 28 candidate apaQTL/eQTL-SNPs were obtained following our request from M.Z. and H.S. (China Nanjing Lung Cancer GWAS), which comprises 8,762 LUAD cases and 13,328 controls from this database. The team utilized this GWAS data have publish articles such as “Identification of risk loci and a polygenic risk score for lung cancer: a large-scale prospective cohort study in Chinese populations”¹ and “Analyses of rare predisposing variants of lung cancer in 6,004 whole genomes in Chinese”².

Currently, the China Nanjing Lung Cancer GWAS data is not publicly accessible, but any organization can apply to this team (M.Z. and H.S.) for this GWAS data. For instance, in the paper “Cross-ancestry genome-wide meta-analysis of 61,047 cases and 947,237 controls identifies new susceptibility loci contributing to lung cancer” published in *Nat Genet.* 2022 Aug;54(8):1167-1177, as the authors mentioned, the GWAS summary statistics of the candidate 45 variants identified from the discovery phase were obtained following their request from M.Z. and H.S. (China Nanjing Lung Cancer GWAS)³.

Your concern about false-positives in our study is very thought-provoking. In GWAS, the *P*-value is used to assess the statistical significance of observed associations between genotypes and phenotypes (such as diseases). Typically, researchers use a significance level to determine which associations are statistically significant. In the past, researchers often set the threshold for significance level to 0.05. However, with

the increase in GWAS data volume and multiple comparisons, this threshold may lead to excessive false positive results. To address this issue, scientists have proposed a more stringent GWAS significance level, typically set at 5×10^{-8} .

However, setting the significance level threshold at 5×10^{-8} may exclude some meaningful results related to diseases since their *P*-values may not reach this strict threshold. The strict *P*-value threshold may overlook some true and causal associations. Generally, adopting a stricter threshold can reduce the risk of false-positives, but it may also increase the likelihood of false negatives and may exclude some potentially important associations. Particularly in cases of small sample sizes or emerging research fields, a stricter threshold may result in missing some true effects.

In our study, using a relative loose *P*-value threshold may help identify some potential causal associations, which may serve as starting points for subsequent more rigorous studies. Based on this significance level criterion, we will design experiments to validate the obtained candidate positive SNPs.

Of course, we quite agree with the reviewer's concern that setting a strict threshold that reaching genome-wide significance can reduce the risk of false-positive, while the identified two SNPs (rs10452178 and rs277646) in our study have not reached genome-wide significance level (5×10^{-8}).

We have mentioned this as one of the limitations in the Discussion as follows: **Setting a strict threshold that reaching genome-wide significance level can reduce the risk of false-positive. Although experiments in vitro and in vivo indicated the causal potential of rs277646 mediating malignant phenotypic changes in LUAD, however, we could not ignore the fact that the rs277646 in China Nanjing Lung Cancer GWAS have not reached genome-wide significance level, which needs further validation in larger sample of Chinese LUAD cases and controls.**

Sorry for the unclear description, during the analysis process, adjustments were made for age, gender, smoking pack-years (if pack-year information is unavailable, then smoking status), and the top 10 principal components to control for potential confounding factors.

Furthermore, thank you for your suggestion regarding whether these SNPs could be identified in large-scale GWAS studies in Europe. Following your suggestion, we have searched large-scale GWAS study based on the European population of the candidate SNPs (Table S2-3). While the lack of significant correlation is disappointing, it is worth noting that these SNPs exhibit the consistent effect directions (OR) in LUAD risk (Table S3). For rs277646, an OR of 1.12 in the Chinese GWAS implies that the G to T mutation increases the risk of LUAD, and similarly, OR values greater than 1 were also found in the European population GWAS databases (ieu-a-965, ieu-a-984), indicating that the G to T mutation may increase the risk of LUAD. For rs10452178, an OR of 0.92 in the Chinese GWAS implies that the C to T mutation decreases the risk of LUAD, and similarly, OR values less than 1 were also found in the European and American population GWAS databases (ieu-a-984), indicating that the C to T mutation may decrease the risk of LUAD. However, it cannot be denied that these results are not significant in European and American populations. Therefore, for greater rigor, we restricted the study population in the title of the article to indicate that it is the Chinese population. The title has been changed to: “**Integrating apaQTL and eQTL analysis identified a potential causal variant associated with lung adenocarcinoma risk in Chinese: a multi-dimensional study**”. We will also add a description of the shortcomings of this article in the discussion section. The description is as follows:

Although our research indicates a potential significant association between candidate SNPs and LUAD in the Chinese population, it's crucial to recognize the limitations of our findings. Features evaluated in European or American populations may be irrelevant or insensitive to the effects of these SNPs.

Typically, larger sample sizes are required to achieve sufficient statistical power, especially when exploring genetic variations that may have moderate effects on phenotypes. Future research efforts should address these limitations by incorporating larger and more diverse sample sizes, considering a broader range of phenotypic features, examining population-specific genetic factors, and exploring gene-environment interactions to gain a more comprehensive understanding of the genetic basis of complex traits in different populations.

- 1 Dai, J. *et al.* Identification of risk loci and a polygenic risk score for lung cancer: a large-scale prospective cohort study in Chinese populations. *Lancet Respir Med* **7**, 881-891, doi:10.1016/S2213-2600(19)30144-4 (2019).
- 2 Wang, C. *et al.* Analyses of rare predisposing variants of lung cancer in 6,004 whole genomes in Chinese. *Cancer Cell* **40**, 1223-1239 e1226, doi:10.1016/j.ccell.2022.08.013 (2022).
- 3 Byun, J. *et al.* Cross-ancestry genome-wide meta-analysis of 61,047 cases and 947,237 controls identifies new susceptibility loci contributing to lung cancer. *Nat Genet* **54**, 1167-1177, doi:10.1038/s41588-022-01115-x (2022).

Table S2. Detailed information on GWAS among the European population

GWAS_ID	Trait	Consortium	Number of SNPs	Population	Case	Control
ieu-a-965	Lung adenocarcinoma	ILCCO	8,881,354	European	3,442	14,894
ieu-a-984	Lung adenocarcinoma	TRICL	10,345,176	European	11,245	54,619

Table S3. The associations between SNPs and LUAD risk

SNP ID	GWAS_ID	Alleles	OR (95%CI)	P -value
rs277646	ieu-a-965	G>T	1.02 (0.90-1.17)	0.750
	ieu-a-984	G>T	1.04 (0.97-1.13)	0.260
	Nanjing Lung Cancer GWAS	G>T	1.12 (1.02-1.22)	0.015
rs10452178	ieu-a-965	NA	NA	NA
	ieu-a-984	C>T	0.97 (0.87-1.08)	0.550
	Nanjing Lung Cancer GWAS	C>T	0.92 (0.87-0.98)	0.009

2. The Nanjing Lung Cancer GWAS database seems to mainly consist of the Chinese Han population. Whereas the public 3'aQTL-atlas and eQTL summary is based on European population data from the GTEx consortium. Due to this difference in population, the minor allele frequency and LD may vary between the two ancestries.

Response: Thank you for your interest in our study and for raising these important questions. Your considerations are valid, as MAF and LD may differ between populations due to ancestral differences. MAF refers to the frequency of the least common allele at a given locus, while LD refers to the non-random association between alleles at two or more loci. These factors are crucial in genomic research as they can influence our interpretation and analysis of genetic variation.

In the section of our study involving SNP analysis related to MAF and LD, we recalculated based on the European population (EUR). Specifically, we employed a criterion of $MAF(EUR) > 0.05$ to obtain 333 SNPs from the initial pool of 338 candidate SNPs. Subsequently, we conducted a consistency assessment between the 333 SNPs obtained from MAF (EUR) and the 256 SNPs obtained from MAF (CHB) (**Figure S1A**). It is worth noting that the 256 SNPs filtered according to the Chinese population's criterion are entirely encompassed within these 333 SNPs, with a concordance rate of 77% (**Figure S1B**). This suggests that the majority of SNPs exhibit similar characteristics across the two populations, enhancing our confidence in the reliability and applicability of these SNPs across different populations.

Figure S1. MAF analysis process and SNP consistency assessment between European and Chinese populations

Further LD analysis in the European population ($r^2 < 0.8$) yielded 33 SNPs. In comparison, screening of the Chinese population resulted in 28 SNPs. Among these, 20 SNPs overlapped, including the positive SNPs (rs10452178, rs11714045, and rs277646) identified after screening a large sample population. This suggests the existence of shared genetic polymorphisms between these two populations. The *Kappa* statistic is a measure used to assess the consistency between two evaluation methods. The range of *Kappa* values is usually from -1 to 1, where a *Kappa* value of 1 indicates complete consistency, that is, perfect consistency between observers or evaluation methods. A *Kappa* value of 0 indicates that the consistency between observers or evaluation methods is the same as randomness, meaning there is no better consistency than randomness. A *Kappa* value less than 0 indicates that the consistency between observers or evaluation methods is lower than the random level, and there may be reverse consistency. In general, a *Kappa* value greater than 0.8 is considered excellent consistency, a value between 0.6 and 0.8 is good consistency, a value between 0.4 and 0.6 is moderate consistency, and a value less than 0.4 is considered poor consistency. Additionally, assessment of the consistency between the two screening criteria using *Kappa* value yielded a *Kappa* value of 0.743, with $P < 0.001$, indicating a high level of agreement between the two screening methods, implying similar conclusions in SNP determination (**Figure S2**). The significance

level of $P < 0.001$ indicates that this inconsistency is not due to random factors but rather has statistical significance, thereby increasing the credibility of this finding.

Figure S2. Consistency heatmap using *Kappa* statistics for LD analysis based on Chinese and European populations

Overall, Although the Minor Allele Frequency (MAF) and Linkage Disequilibrium (LD) exhibit high consistency between Chinese and European populations, there are indeed differences in the frequency of secondary alleles and LD between the Chinese and European populations, and we will add a description of the limitations in the discussion of the article:

When discussing the limitations of this study, it's important to acknowledge the influence of genetic heterogeneity within and between populations. Despite the Nanjing Lung Cancer GWAS database appearing to be predominantly composed of the Chinese Han population, the public 3'aQTL-atlas and eQTL summary is based on European population data from the GTEx consortium. Due to these population differences, MAF and LD may vary between the two ancestries, potentially affecting the results of GWAS studies. In subsequent research,

consideration of the impact of racial differences may also be necessary.

3. In “Table 1 Detail information of the 28 candidate apaQTL/eQTL-SNPs”, it seems that there are multiple SNPs associated with each gene. However, is it possible to determine whether their effect sizes are in the same direction? Furthermore, if each SNP has a significant, positive or negative effect on a gene’s molecular phenotype (such as expression or APA), how can we identify the causal SNP?

Response: Thank you for your interest in our research and for raising these important questions. Please allow me to provide explanations regarding these questions.

Firstly, it’s true that each gene indeed contains multiple SNPs, and each SNP has the potential to influence gene function. The consistency effect observed in our study implies uniformity in gene expression, hence we anticipate a negative correlation between APA results and eQTL results. That is, mutations in APA-associated SNPs leading to an increase in PDUI values tend to favor the use of distal poly(A) sites, resulting in increased binding of miRNAs and RBPs, consequently leading to decreased gene expression, with the corresponding eQTL results showing a declining trend. Conversely, a decrease in PDUI values tends to favor the use of proximal poly(A) sites, allowing miRNAs and RBPs to escape, corresponding to an increase in gene expression, with eQTL results showing an upward trend for the respective SNP.

In this study, the APA and eQTL effects of three SNPs (rs277646, rs1214375, rs11714045) on *NIT2* and two SNPs (rs10452178, rs223332) on *CISD2* were consistent (**Table S4, Figure S3**). When the APA effect shows a decrease in PDUI value, the eQTL effect leads to an upregulation of gene expression; On the contrary, when the APA effect shows an increase in PDUI value, the eQTL effect leads to downregulation of gene expression.

Table S4. Effect information of candidate apaQTL/ eQTL-SNPs

Gene	SNP	allele	eQTL (Gene expression)	apaQTL (PDUJ)	OR (95%CI)	P value
CISD2	rs10452178	C > T	Down↓	Up↑	0.92(0.87-0.98)	0.009
	rs223332	G > T	Up↑	Down↓	1.00(0.96-1.05)	0.923
NIT2	rs277646	G > T	Up↑	Down↓	1.12(1.02-1.22)	0.015
	rs1214375	G > A	Down↓	Up↑	0.98(0.94-1.03)	0.522
	rs11714045	C > T	Down↓	Up↑	1.05(1.00-1.10)	0.076

Figure S3. EQTL and APA effect map of candidate apaQTL/eQTL SNPs

Specifically, for rs277646 on *NIT2*, compared to the G genotype, the T genotype exhibited a decrease in PDUJ value for the APA effect, resulting in an eQTL effect of upregulating gene expression (**Figure S4**).

Figure S4. The Relationship between Different Genotypes of rs277646 and eQTL and APA Effects

For the candidate SNPs, we confirmed causal SNPs through human and cell model experiments. We conducted large-scale population studies to validate the association between these SNPs and LUAD. Our analysis revealed significant associations between some SNPs and LUAD, further supporting their potential involvement in the occurrence and progression of LUAD. Our study aimed to identify SNPs with eQTL (expression quantitative trait loci) and apaQTL (alternative polyadenylation quantitative trait loci) functionalities and investigate their relationship with LUAD.

Specifically, eQTL analysis of SNPs was performed on the GTEx website (<https://gtexportal.org/>). For example, SNP (rs277646) showed an increased in *NIT2* gene expression after the G > T mutation. However, APA analysis of SNPs was conducted using the 3'aQTL-atlas website (<https://wlcboit.uci.edu/3aQTLatlas/>),

revealing that after the rs277646 G > T mutation, the PDUI value of *NIT2* decreased, the proportion of usage of the 3'UTR distal site decreased, the proximal polyadenylation (a) sites were more likely to be used, leading to a decrease in miRNA and RBP binding and an increase in post-transcriptional gene expression. Previous biogenic analysis showed that *NIT2* is highly expressed in LUAD, so increased *NIT2* gene expression after mutation would logically lead to an increased risk of LUAD (OR>1), and our large sample of GWAS showed consistent results. Similarly, the biological validity of rs10452178 on *CISD2* is consistent with the performance of GWAS in large samples (Figure S5).

Figure S5. Schematic diagram of changes in LUAD risk after apaQTL-SNP-mediated mutation

For SNPs that exhibited logical consistency, we conducted further cellular and animal experiments to gain a deeper understanding of their causal relationship with LUAD.

Through these experiments, we were able to observe whether mutations in candidate SNPs affected the onset of LUAD, further validating their role in LUAD occurrence. We believe that our approach, combining computational analysis with experimental validation, provides a robust framework for elucidating the functional relevance of identified SNPs in the context of LUAD.

4. The finding of the intronic variant rs277646 effect on 3'UTR length is interesting, but the explanation of the mechanism is quite descriptive and superficial. The authors need to conduct experiments to confirm which RBP/microRNA was indeed involved in this interaction. Without this evidence, it is difficult to evaluate the significance of this finding.

Response: Thank you for providing further details on our research findings. The 3'RACE experiments detected selective recognition of the PAS1 and PAS3 sites with polyadenylation in the *NIT2* gene, occurring at positions 949bp (PAS1) and 1203bp (PAS3), respectively. This suggests that the *NIT2* gene selectively recognizes the PAS1 and PAS3 sites. To validate the involvement of microRNAs in this interaction, we firstly predicted which microRNAs can bind between PAS1 and PAS3 using the ENCORI database (<http://starbase.sysu.edu.cn/>) and miRDB (<https://mirdb.org/ontology.html>). Ultimately, we identified hsa-miR-650 and hsa-miR-642a-3p may bind between PAS1 and PAS3.

Specifically, PAS1 is located 83bp downstream from the start site of the *NIT2* 3'UTR, while PAS3 is situated 337bp downstream from the start site of the *NIT2* 3'UTR. hsa-miR-650 binds to the region spanning 248-255bp of the *NIT2* 3'UTR, whereas hsa-miR-642a-3p binds to the region spanning 320-326bp of the *NIT2* 3'UTR (https://www.targetscan.org/vert_80/) (**Figure S6A**).

Figure S6. Mapping miRNA Binding Patterns and Interactions in the *NIT2* 3'UTR Region

Subsequently, we designed a dual luciferase reporter gene experiment to verify the binding of miRNAs (hsa-miR-650 and hsa-miR-642a-3p) on the 3'UTR of the *NIT2*. The luciferase reporter plasmids involved in the assay were constructed by Jingrui (Guangzhou, China). The results showed that the hsa-miR-650 mimics or hsa-miR-642a-3p mimics did not bind to the short 3'UTR of *NIT2* (~PAS1). However, in the long 3'UTR of *NIT2* (~PAS3), the has-miR-650 mimics significantly reduced luciferase activity, indicating that the has-miR-650 mimics binds to the long 3'UTR of

NIT2 (~PAS3); The luciferase activity of hsa-miR-642a-3p mimics did not decrease, indicating that hsa-miR-642a-3p mimics does not bind to the long 3'UTR of *NIT2* (~PAS3). Therefore, miR-650 is indeed involved in the interaction between this gene and miRNA (**Figure S6B**). The rs277646-T genotype causes *NIT2* to preferentially utilize the proximal poly (A) site, resulting in a shorter 3'UTR transcript. This leads to the loss of the hsa-miR-650 binding site on *NIT2*, thereby affecting the expression level of *NIT2*. The specific content has been described in detail in the corresponding section of the article.

METHODS

Fluorescent Enzyme Reporter Gene Assay

Fluorescent enzyme activity was measured using the Dual-Luciferase Reporter Assay System (Promega). According to the manufacturer's instructions, 293T cells with a transfection efficiency of 40% were transfected with the transfection reagent Fitran (HuiJun, Guangzhou, China) in a 24-well plate. The psiCHECK2 vector was obtained from Jingrui (Guangzhou, China). Transfection groups were as follows: (i) Blank control (transfection reagent only), (ii) psiCHECK2(empty vector) + miRNA NC, (iii) psiCHECK2(empty vector) + hsa-miR-650 mimics, (iv) psiCHECK2(empty vector) + hsa-miR-642a-3p mimics, (v) psiCHECK2-PAS1 + miRNA NC, (vi) psiCHECK2-PAS1 + hsa-miR-650 mimics, (vii) psiCHECK2-PAS1 + hsa-miR-642a-3p mimics, (viii) psiCHECK2-PAS3 + miRNA NC, (ix) psiCHECK2-PAS3 + hsa-miR-650 mimics, (x) psiCHECK2-PAS3 + hsa-miR-642a-3p mimics. Cells were collected 48 hours post-transfection, and the Dual-Luciferase Reporter Assay System (Promega, E1910) was used for analysis.

RESULTS

The impact of hsa-miR-650 binding to the long 3'UTR of *NIT2*

Due to the different genotypes of rs277646 affecting the expression of different subtypes of *NIT2*, we further investigated its influence on gene-miRNA interactions. Firstly, we selected microRNAs using the ENCORI database

(<http://starbase.sysu.edu.cn/>) and miRDB (<https://mirdb.org/ontology.html>), ultimately identifying hsa-miR-650 and hsa-miR-642a-3p as two microRNAs. According to the prediction results from the TargetScan website (https://www.targetscan.org/vert_80/), both microRNAs are predicted to bind between Poly(A)1 and Poly(A)3. Specifically, Poly(A)1 is located 83bp downstream from the start site of the *NIT2* 3'UTR, while Poly(A)3 is situated 337bp downstream from the start site of the *NIT2* 3'UTR. hsa-miR-650 binds to the region spanning 248-255bp of the *NIT2* 3'UTR, whereas hsa-miR-642a-3p binds to the region spanning 320-326bp of the *NIT2* 3'UTR (**Figure 6H**). To validate the involvement of microRNAs in this interaction, we constructed short sequences (~PAS1) and long sequences (~PAS3) for the luciferase reporter gene assay.

Results from the luciferase reporter gene assay showed that the hsa-miR-650 mimics or hsa-miR-642a-3p mimics did not bind to the short 3'UTR of the *NIT2* (~PAS1). However, in the long 3'UTR of *NIT2* (~PAS3), the has-miR-650 mimics significantly reduced luciferase activity, indicating that the has-miR-650 mimics binds to the long 3'UTR of *NIT2* (~PAS3); The luciferase activity of hsa-miR-642a-3p mimics did not decrease, indicating that hsa-miR-642a-3p mimics does not bind to the long 3'UTR of *NIT2* (~PAS3). (**Figure 6I**). Therefore, the rs277646-T genotype leads to *NIT2* preferentially utilizing the proximal poly(A) site, resulting in shorter 3'UTR transcripts, which leads to the loss of hsa-miR-650 binding sites on *NIT2*, thereby affecting the expression levels of *NIT2*.

Minor:

1. lack reference.

-Line 56," Lung adenocarcinoma (LUAD) and lung squamous cell carcinoma (LUSC) have shown distinct incidence trends in recent decades..." lack reference.

-Lines 61-63 lack reference.

Response: Dear reviewer, thank you for your question.

For line 56, we have included the relevant references in the revised manuscript^{4,5}.

For 61-63 lines of evidence from GWAS catalog website

(<https://www.ebi.ac.uk/gwas/>), we are added in the revised version, is as follows:

“The evidence from GWAS Catalog (<https://www.ebi.ac.uk/gwas/>) showed that the currently reported LUAD susceptibility regions are far more than that of LUSC, indicating that genetic factors may affect the susceptibility of LUAD in a large extent.”

- 4 Siegel, R. L., Miller, K. D. & Jemal, A. Cancer statistics, 2020. *CA Cancer J Clin* **70**, 7-30, doi:10.3322/caac.21590 (2020).
- 5 Sung, H. *et al.* Global Cancer Statistics 2020: GLOBOCAN Estimates of Incidence and Mortality Worldwide for 36 Cancers in 185 Countries. *CA Cancer J Clin* **71**, 209-249, doi:10.3322/caac.21660 (2021).

2. The expression of *NIT2* in LUAD cell lines A549 was not significantly higher than that of the control in Figure 4A.

Response: Dear reviewer, thank you for pointing out this issue, you are absolutely correct. We overlooked this important detail, and have now made the necessary revisions to the manuscript. The amended content is as follows:

“The results indicate that NIT2 expression in LUAD cell lines (PC9 and SPCA1) is significantly higher than in HBE cells. Additionally, there is a trend of higher expression of NIT2 in the LUAD cell line A549 compared to HBE cells.”

3. The method section “Study population of the susceptibility study” is confusing; what is a susceptibility study? There's not much mentioned in the main text about the results of this section.

Response: Thank you, dear reviewer, for the reminder. We apologize for the lack of clarity in the description of the method section. Susceptibility research is a research method that focuses on the degree of response or disease risk of individuals in specific environments, lifestyles, genes, and other factors. Our study aims to explore the association between SNPs (single nucleotide polymorphisms) with apaQTL and eQTL functions and susceptibility to LUAD in the Chinese population. SNPs are one of the common forms of variation in the human genome, which may affect an individual's

susceptibility to diseases. Through this study, we aim to reveal the potential link between specific SNPs in the Chinese population and the risk of LUAD, in order to provide more precise guidance and personalized medical strategies for prevention, diagnosis, and treatment.

In response to the reviewer's queries, I have provided a detailed description in the methods section. The description is as follows:

Susceptibility research

Genetic susceptibility research is typically conducted in large-scale populations, employing genome-wide association studies (GWAS). These studies compare genetic variations (single nucleotide polymorphisms, SNPs) among individuals to explore their relationship with specific diseases or traits. For the included potential functional candidate SNPs, we employed logistic regression analysis to assess the association between apaQTL/eQTL SNPs and the risk of LUAD. We computed results based on odds ratios (OR) and 95% confidence intervals (CI) to quantify the degree of association between SNPs and the risk of LUAD. In the analysis process, adjustments were made for age, gender, smoking pack-years (or smoking status if pack-year information was unavailable), and the top 10 principal components to control for these potential confounding factors. The frequency distribution of positive SNP genotypes conforms to Hardy Weinberg's equilibrium ($P>0.05$), indicating that the selected population has sufficient representativeness.

4. Typo error:

-Spearman's rank correlation coefficient is often denoted by the Greek letter ρ (rho) or r_s (italic, please).

-Gene names should be in italics.

-3'UTR, 3' RACE.

Response: Thank you very much for discovering this error. We apologize for this this typo issue and corrected it based on your suggestions.

Reviewer #2

Xu et al aimed to investigate the association between the risk of LUAD and apaQTL/eQTL-SNPs. APA-related LUAD genes were screened, and candidate SNPs with both apaQTL and eQTL functions were identified. Large-scale GWAS analysis confirmed significant associations between rs10452178, rs277646 alleles, and LUAD risk. Functional experiments validated that the rs277646 variant T allele was associated with increased *NIT2* expression, shorter 3'UTR transcript length, and promoted malignant phenotypes and tumor growth in LUAD. These findings reveal the potential impact of apaQTL/eQTL-SNPs on LUAD and provide new insights into the pathogenesis of LUAD. The discoveries in this study may contribute to the early risk assessment and treatment strategies for LUAD.

Comment 1: The design was not perfect when selecting APA related genes. The criteria of $|R_s| > 0.3$ and $FDR < 0.05$ were used in the selection of APA-related genes. These thresholds may be too loose in the definition of genes associated with APA events, which may result in too many genes being included in the study, potentially affecting the accuracy and interpretation of the results.

Response: Dear reviewer, thank you very much for providing feedback on the statistical criteria we have chosen. We acknowledge that $|R_s|$ is not super relevant on a 0.3 scale and we highly value your input and would like to provide further clarification on the points you raised.

In general, 0.3 is typically considered a moderate strength correlation coefficient. This indicates that there is a certain level of correlation between two variables, although not very strong, it is also not weak. Therefore, when we observe a Spearman correlation coefficient greater than 0.3, we do not consider this level of correlation low.

In specific contexts, 0.3 may be regarded as an acceptable level of correlation, especially in certain social science fields or medical research, where this level of

correlation already holds practical significance and may be widely accepted and referenced. For example, Lan et al. used a Spearman rank correlation coefficient $|R_s| > 0.3$ as a correlation criterion in their integrated analysis of multi-omics data⁶. Li et al., while exploring lncRNA relevant to potential clinical applications, considered a Spearman rank correlation coefficient (absolute value) of 0.3 as significant correlation⁷. In the study by Nienke M ter Haar et al., the Spearman rank test with $|R_s| = 0.1-0.3$ was considered weak, $|R_s| = 0.3-0.5$ as moderate, and $|R_s| > 0.5$ as strong test criteria⁸.

Furthermore, regarding our adoption of the criteria $|R_s| > 0.3$ and $P_{FDR} < 0.05$, it is because this portion of our data is derived from Xiang et al.'s previous research, and thus we adhere to their standards. In their characterization of global selection of adenylated events (APA) across different cancer types in The Cancer Genome Atlas (TCGA) database, genes associated with APA events were defined using $|R_s| > 0.3$ and P_{FDR} less than 0.05 (reference #14 in the revised manuscript).

It is worth noting that in our previous study, we found that the concordance between differentially expressed mRNAs and proteins is only approximately 20-40%. This underscores the importance of integrating transcriptomic and proteomic data (**Figure S7**)⁹. Therefore, we further assessed whether there were differential expressions at the corresponding protein level. Although our initial inclusion criteria were relevant loose, we subsequently rigorously filtered based on consistent expression between proteins and mRNA.

Figure S7. Consistency rate of differential mRNA and protein expression in the Chinese population

Specifically, we further analyzed the expression differences of the aforementioned 518 genes at the mRNA level between 57 pairs of LUAD tumor tissues and adjacent non-tumor tissues using the TCGA database. The results revealed that a total of 143 genes exhibited differential expression at the mRNA level ($|FC| > 1.5$, $P < 0.05$). Subsequently, we validated these 143 genes between 49 pairs of LUAD tumor tissues and adjacent non-tumor tissues from the Chinese population, among which 65 genes were confirmed to be significantly differentially expressed ($|FC| > 1.5$, $P < 0.05$). The results revealed 32 proteins showing differential expression between tumor tissues and adjacent non-tumor tissues from the same Chinese population, and the expression directions of these 32 proteins were consistent with their corresponding mRNAs (**Figure S8**). Consequently, we conducted further investigation on the overlapping 32 APA-related LUAD genes, which exhibited consistent differential expressions at both mRNA and protein levels. This supplementary step may help address any limitations

of the initially loose criteria and enhances the accuracy of our results.

Figure S8. Schematic representation the selection of APA-related genes

We acknowledge that the criteria of $|R_s| > 0.3$ and $P_{FDR} < 0.05$ are relatively loose, potentially may impact the accuracy of the results, for which we sincerely apologize. Therefore, we have added a discussion on the limitations of the study as follows: **“Although we rigorously filtered based on consistent gene expression patterns both between protein level and mRNA level, when selecting APA-related genes, using $|R_s| > 0.3$ and $P_{FDR} < 0.05$ as criteria for defining genes related to APA events might be loose, thus potentially compromising the accuracy and interpretation of the results.”**

- 6 Lan, Y. *et al.* AtMAD: Arabidopsis thaliana multi-omics association database. *Nucleic Acids Res* **49**, D1445-D1451, doi:10.1093/nar/gkaa1042 (2021).
- 7 Li, J. *et al.* TANRIC: An Interactive Open Platform to Explore the Function of lncRNAs in Cancer. *Cancer Res* **75**, 3728-3737, doi:10.1158/0008-5472.CAN-15-0273 (2015).
- 8 Ter Haar, N. M. *et al.* In silico validation of the Autoinflammatory Disease Damage Index. *Ann Rheum Dis* **77**, 1599-1605, doi:10.1136/annrheumdis-2018-213725 (2018).

- 9 Liu, Y. *et al.* Identification of the consistently differential expressed hub mRNAs and proteins in lung adenocarcinoma and construction of the prognostic signature: a multidimensional analysis. *Int J Surg* **110**, 1052-1067, doi:10.1097/JS9.0000000000000943 (2024).

Comment 2: In this study, a particular concern was whether the correlation coefficients observed in cancer samples (line 315) reached a threshold that is generally considered significant in scientific studies. In the study, a correlation coefficient of about 0.3 is considered a low correlation level. Therefore, if the correlation coefficient in the cancer sample is lower than this value, there may be a risk that the association is not significant enough, which may require further consideration and validation. In scientific research, it is essential to ensure the rigor of data analysis and the reliability of results. Therefore, this factor should be carefully considered when interpreting and inferring these results.

Response:

Dear reviewer, thank you for your reminder, and we sincerely apologize for the oversight in this section. The situation at line 315, where the correlation with gene expression fell below 0.3, occurred because we used FPKM (Fragments Per Kilobase Million) data units to measure gene expression levels. This was indeed an oversight on our part, and we appreciate your clarification.

In fact, the APA-related genes we obtained are derived from the study conducted by Xiang et al. They noted in their research: **“To avoid the potential batch effects introduced by gene expression quantification, we recalculated gene expression as reads per kilobase per million mapped reads (RPKM) across all TCGA and CCLE samples.”** Therefore, in their research, they employed the criterion of $|Rs| > 0.3$, which is based on RPFM (Reads Per Kilobase Million) data units¹⁰. While FPKM and RPFM calculations are similar, there are slight differences in the factors considered in the calculation formula. RPFM not only accounts for gene length, thus correcting for differences in expression levels among genes of varying lengths, but also corrects for differences in sequencing depth among different samples. This

renders RPKM more comparable when assessing gene expression levels across different samples.

Due to an oversight in data unit usage, we recalculated the data. The results show that all the $|R_s|$ values exceed 0.3 in total tissues and adjacent non-tumor tissues as well as LUAD tumor tissues. In revised **Figure 2A-F**, we incorporated relevant visualizations. As shown in **Figure 2A-C**, in adjacent non-tumor tissues ($R_s = -0.540$, $P = 1.23 \times 10^{-5}$), LUAD tumor tissues ($R_s = -0.407$, $P = 9.50 \times 10^{-21}$), and total tissues ($R_s = -0.500$, $P = 1.28 \times 10^{-36}$), we all observed a negative correlation between the PDUI value and gene expression level of *CISD2*. As the PDUI value increased, the expression of *CISD2* decreased. In addition, the PDUI value of *NIT2* was also negatively correlated with *NIT2* gene expression level in both adjacent non-tumor tissues ($R_s = -0.371$, $P = 4.12 \times 10^{-3}$), LUAD tumor tissues ($R_s = -0.320$, $P = 5.60 \times 10^{-13}$), and total tissues ($R_s = -0.361$, $P = 3.82 \times 10^{-18}$) (**Figure 2D-F**).

We appreciate your constructive feedback once again and assure you that we will carefully review all data and unit usage to ensure accuracy and consistency. We will include the following statement in the Methods section: "To avoid potential batch effects from gene expression quantification, gene expression was recalculated as reads per kilobase per million mapped reads (RPKM) for all TCGA samples"

when calculating the correlation between gene expression and PDUI value."

We greatly appreciate your meticulous approach to research. Indeed, while we recalculated the $|R_s|$ value using RPKM, it cannot be denied that in the analysis of the correlation between *NIT2* and PDUI in LUAD tumor tissues, the $|R_s|$ is 0.320, slightly exceeding 0.3. This suggests a potential risk of insufficient correlation significance, which may warrant further consideration and validation. Therefore, this factor should be carefully considered when interpreting and inferring these results.

Based on these results, we have added a limitation to our discussion: **"The correlation between PDUI and *NIT2* expression in LUAD tumor tissues is moderate, with a $|R_s|$ value of 0.320, slightly exceeding 0.3, suggesting a potential risk of insufficient correlation significance, which may warrant further consideration and validation. Therefore, this factor should be carefully considered when interpreting and inferring these results."**

10 Xiang, Y. *et al.* Comprehensive Characterization of Alternative Polyadenylation in Human Cancer. *J Natl Cancer Inst* **110**, 379-389, doi:10.1093/jnci/djx223 (2018).

Reviewer #3

Xu et al. integrated APA-related LUAD genes according to the consistently differential expression both at the mRNA and protein levels, and then mapped apaQTL/eQTL-SNPs from 3'aQTL-atlas to APA-related LUAD genes. In the following susceptibility study, they evaluated the association between selected apaQTL/eQTL-SNPs and LUAD risk through 8,762 LUAD cases and 13,328 cancer-free healthy controls. The authors conclude that the variant T allele of rs277646 is associated with shortened 3'UTR length of *NIT2* mRNA and may increase the risk of developing LUAD under the evidence that the variant T allele of rs277646 promoted malignant phenotypes in LUAD cell line and tumor growth in mice. Overall, this study is innovatively designed and carefully carried out, which may provide novel clues for understanding and exploring APA events in LUAD carcinogenesis. The

manuscript would benefit from some additional information and please see comments below:

Comment 1: Legends should be short stand-alone described that allows the readers to understand the Figures.

Response: Dear reviewer, we sincerely appreciate your invaluable feedback. We have made optimizations to some of the figure legends in the manuscript as per your suggestions.

Comment 2: In Figure 2, how the PDUI values were normalized?

Response: Thank you for your kind reminder, dear reviewer. We apologize for the lack of clarity in the description of our data processing.

The normalization of PDUI for **Figures 2A-F** is explained as follows: We have performed batch normalization correction on RNA-seq data from various databases. We have added a normalization description for PDUI in the Methods section of the manuscript as follows: **“Due to the different expression ranges of genes in various databases, normalization of the data was necessary for reliable and meaningful comparisons. We balanced these values through a min-max normalization process, which transformed them into a range of (0,1).”**

The normalization of PDUI for figures 2G-I is explained as follows: This portion of PDUI data in relation to SNPs is sourced from the research conducted by the Cui et al. They previously developed DaPars2 for calculating the relative usage of APA, which employs a bimodal Gaussian mixture model, enabling joint quantification of APA usage across multiple samples. Sequencing depth for each sample is computed and utilized as input for DaPars2. The sequencing depth differences across samples are normalized using SAMtools v1.9.

Comment 3: There is nothing called borderline significant. It should be revised.

Response: Dear reviewer, thank you for your feedback regarding the term “borderline significant.” We understand your concern and appreciate the opportunity to address it. We acknowledge that the term “borderline significant” may not be standard in statistical terminology and could potentially lead to ambiguity. Upon reflection, we agree that clarity and precision in reporting statistical findings are paramount. To address this concern, we will revise the terminology in our manuscript. Instead of using “borderline significant”, we will employ terms such as “approaching significance” or “nearing significance” to better convey the proximity of our results to conventional significance levels without implying a definitive statistical outcome. We appreciate your diligence in reviewing our manuscript and your valuable feedback. Please let us know if you have any further suggestions or concerns.

Comment 4: It is valuable that the authors considered the low consistency rate of differentially expressed mRNAs and its’ coding proteins, what is the consistency rate in this study? Are the results similar to other studies?

Response: Thank you, dear reviewer, for the reminder. We apologize for the lack of clarity in the description of the data processing. In our study, based on APA-related genes in LUAD, we combined the TCGA database to further determine whether they were also differentially expressed in 57 paired LUAD tumor tissue and adjacent non-tumor tissues, obtaining 143 genes (mRNA levels). We then validated that these genes were also differentially expressed using a comprehensive transcriptomics database of LUAD (49 pairs of LUAD tumor tissues and adjacent non-tumor tissues) from the Chinese population (GSE140343), obtaining 88 differentially expressed genes ($P < 0.05$).

Subsequently, 143 genes (mRNA levels) were combined with the above proteome database of the same Chinese population (IPX0001804000) to obtain 89 LUAD-related genes (protein levels) that were also differently expressed at their encoded protein levels ($P < 0.05$). By setting fold changes (FC)⁺ with $|FC| > 1.5$, 65 differentially expressed genes (mRNA levels) and 69 differentially expressed genes

(protein levels) were obtained. Finally, 32 overlapping APA-related LUAD genes with consistent differential expression at mRNA and its encoded protein levels were obtained (**Figure S9**).

Figure S9. Concordant Expression Overlap Venn Diagram

Based on this, the concordance rate for differentially expressed mRNAs and their encoded proteins was calculated to be 35.96% (32/89).

In our previous study⁹, the consistency between differentially expressed messenger RNA and protein was approximately 20-40%, which is consistent with previous research.

- 9 Liu, Y. *et al.* Identification of the consistently differential expressed hub mRNAs and proteins in lung adenocarcinoma and construction of the prognostic signature: a multidimensional analysis. *Int J Surg* **110**, 1052-1067, doi:10.1097/JS9.0000000000000943 (2024).

Comment 5: In Figure 4A. The results showed that compared with HBE cell lines, NIT2 was overexpressed in the PC9 and SPCA1 cells. Why only SPCA1 was chosen to construct the mutant cell instead of the PC9?

Response: Dear reviewer, thank you for your questions about **Figure 4A**. We appreciate your attention to detail and would like to clarify the selection of SPCA1

over PC9 to build mutant cells.

In preliminary experiments, we separately transfected the mutant plasmids into both SPCA1 and PC9 cells and assessed transfection efficiency using fluorescence microscopy. This method allowed us to observe the intensity and efficiency of fluorescence under the microscope. Through our experimental observations, we found that SPCA1 exhibited significantly higher transfection efficiency, as evidenced by the strong fluorescence signal observed in SPCA1-transfected cells. In contrast, PC9 cells showed relatively lower transfection rates and weaker fluorescence signals.

The disparity in transfection efficiency between SPCA1 and PC9 led us to choose SPCA1 as our preferred cell line for the study. Considering the importance of efficient transfection in obtaining reliable experimental results, we believe that using SPCA1 enables us to acquire more accurate and reproducible data.

Comment 6: In the study, the authors suggested that the *NIT2* rs277646 variant T allele is associated with shortened 3'UTR length of *NIT2* mRNA and may increase its expression hence may affect malignant phenotypes in LUAD cell line and tumor growth in mice. In Figure 4B, the authors successfully obtained *NIT2*-rs277646-G through CRISPR/Cas9 mediated genome editing. Does the mutation of *NIT2*-rs277646 from T to G affect *NIT2* expression? These data are missing.

Response: Thank you, dear reviewer, for the reminder. We sincerely apologize for the lack of this portion of the data. To address this gap, we have conducted a western blot experiment to comprehensively describe the impact of the T to G mutation of *NIT2*-rs277646 on *NIT2* expression post-cell transfection. Additionally, we have supplemented the description in the section regarding the “Effects of *NIT2*-rs277646 on the malignant phenotype of LUAD in vitro and vivo” as follows:

“Meanwhile, Western blot analysis revealed that the expression of *NIT2* in the *NIT2*-rs277646-T was higher than that in the *NIT2*-rs277646-G ($P = 0.0180$) (Figure 4C).”

REVIEWERS' COMMENTS:

Reviewer #1 (Remarks to the Author):

I remain unconvinced by the authors' response. They have examined the mechanisms of two SNPs with adjusted p-values that are far from significant, which are associated with lung cancer in the Chinese population. Their new data also clearly demonstrate that these risk SNPs are not significant in any related studies involving European populations. This raises questions about the rationale behind selecting these non-significant SNPs for mechanistic studies and for furthering our understanding of lung cancer carcinogenesis. Additionally, all the mechanistic data used in this study, such as the 3'aQTL-atlas and eQTL summary, are based on different ancestries. Moreover, the explanation of the mechanisms remains descriptive and superficial.

Reviewer #2 (Remarks to the Author):

The authors have addressed all my concerns, and I recommend to accept this manuscript.

Reviewer #3 (Remarks to the Author):

All my concerns were well addressed.